# FROM REGRESSION TO DOSE–RESPONSE: A FRAMEWORK TO PREDICT ACTIVITY AND $EC_{50}$ FOR GPCRs

## ABSTRACT

ML models have revolutionised structural biology and significantly advanced drug discovery, yet they struggle with predicting the ligand-induced activity of G-protein coupled receptors (GPCRs). GPCR proteins are cellular membrane "sensors", which trigger a cascade of intracellular processes upon binding a diverse set of molecules. Human GPCRs account for nearly $30\%$ of targets of approved drugs, and approximately half of them are olfactory receptors (ORs). Beyond their role in smell perception, ORs are increasingly linked to diseases such as obesity, diabetes, asthma, and cancer. The core interest and difficulty in modelling the molecule-induced response of ORs and GPCRs lie in predicting activity and potency (i.e. half maximal effective concentration, $EC_{50}$). In this paper, we propose a new way of modelling these properties. Instead of direct regression on $EC_{50}$ values, we mimic *in vitro* dose-response assays by sampling binary activity labels for a protein-molecule pair $(s, m)$ at a molecular concentration $c$. Then we design a novel model that learns the activation probability $P(active|s, m, c)$ at any given $c$. Finally, querying the model across concentrations enables fitting a logistic curve, from which both activity (curve maximum) and $EC_{50}$ (inflection point) are derived. On a challenging M2OR dataset, our framework improves activity prediction by $10\%$ over the state-of-the-art. For $EC_{50}$ estimation, it achieves an error of $0.725$ log units, $40\%$ lower compared to a regression baseline, and surpasses the affinity module of Boltz-2 by $0.385$ log units. Notably, our approach effectively identifies novel active scaffolds, demonstrating its potential to replace expensive *in vitro* primary screening. The proposed framework is protein-agnostic, and we observe state-of-the-art performance in estimating activity and dissociation constant ($K_d$) on standard drug-target affinity benchmarks, DAVIS and BinidngDB.

## 1 INTRODUCTION

Machine learning models in protein biology (Jumper et al., 2021; Watson et al., 2023; Passaro et al., 2025) have brought unprecedented level of exploration, significantly speeding up drug discovery pipelines (Du et al., 2024). Among protein families, G-protein coupled receptors (GPCRs) hold a prominent place as highly important drug targets due to their role in transmitting the chemical signals from the external environment to the cell. However, modelling the molecule-induced activity of these proteins is notoriously difficult as subtle differences in molecular structure can significantly alter protein-ligand interactions, even transforming an agonist into an inverse agonist (Kosar et al., 2024; Qin et al., 2022). Similarly, a single point mutation can alter the activity (de March et al., 2018). All these small variations can drastically change the cellular response, leading to differences in efficacy (the highest response) and $EC_{50}$ (the concentration needed to achieve $50\%$ of the highest response) (Figure 1) (Heydenreich et al., 2023). In particular, accurate estimation of $EC_{50}$ is crucial in determining drug dosage and avoiding off-target responses (Zhang et al., 2024).

The largest subfamily of GPCRs are olfactory receptors (ORs), which constitute $49\%$ of all genes encoding GPCRs (Bjarnadóttir et al., 2006; Niimura & Nei, 2003). For a given odourant, the specific pattern of activated ORs serves as a signature that encodes its odour identity (Malnic et al., 1999; Nara et al., 2011) and ORs with a lower $EC_{50}$ appear to have a greater importance in this signature (Junek et al., 2010; Spors & Grinvald, 2002; Wilson et al., 2017). Beyond their role in olfaction, these receptors have a widespread presence throughout the body. After the first report of their

expression in testes (Parmentier et al., 1992), OR transcripts have been found in various tissues, including heart, kidney, liver, lungs, prostate, brain, or leukemia cells. Thence, ORs hold a promise for potential therapeutic and diagnostic applications in diseases such as asthma, obesity, diabetes, and cancer (Lee et al., 2019).

Currently, the method of choice to characterise molecule-induced OR activity are *in vitro* experiments, which consist of preliminary screening rounds followed by a detailed dose-response assay that allows estimating $EC_{50}$ and efficacy. However, despite the progress in the throughput of *in vitro* assays driven by engineered heterologous systems (Saito et al., 2004; 2009), available data covers only a small fraction of the millions of possible combinations between molecules and receptors. The most comprehensive database of *in vitro* experiments (Lalis et al., 2024a) lists 6157 dose-response assays, of which only 1663 observe a cellular response and estimate $EC_{50}$. Thus, the best bet to reveal the activity of OR-molecule pairs are *in silico* approaches and a model capable of predicting $EC_{50}$ for any given pair is essential to fully characterise the odour coding and drugability of ORs.

In this work, we confront the standard paradigm of separate $EC_{50}$ and activity prediction tasks. We draw an analogy to how the dose-response curves are fitted *in vitro*, and instead of treating $EC_{50}$ prediction as a regression, we model the underlying biological experiment. In dose-response assays, the response of a protein-molecule pair is measured in several molecular concentrations. The activity is then assessed by the highest response, and the $EC_{50}$ is estimated by fitting a generalised logistic curve to these measurements. In analogy, we design a model that predicts the probability of activation for a protein-molecule pair at a given molecular concentration. By querying this model at several concentrations – mimicking an experimental dose-response assay – we can then fit a logistic curve to these predictions. This unified framework yields both the final activation decision (the curve's maximum) and the estimated $EC_{50}$ (the curve's inflection point).

Following this approach, we outperform the current state-of-the-art in activity decision task by $10\%$ on a challenging M2OR dataset. Strikingly, by decomposing the difficult regression problem into a series of simpler binary classifications, our approach reduces the $EC_{50}$ estimation error by $40\%$ compared to a traditional regression baseline. In addition, it outperforms *in vitro* screening campaigns in predicting the response of protein variants and novel molecules within the training chemical space, while outperforming primary screening even in the challenging search for new active scaffolds. The proposed framework, primarily designed for GPCRs, also demonstrates high performance in dissociation constant ($K_d$) estimation for other types of proteins. On two drug-target affinity datasets, our approach achieves state-of-the-art $K_d$ estimation error while increasing the activity decision performance by $7\%$ and $3\%$ compared to models specifically designed for kinase inhibition.

## 2 RELATED WORK

Despite the potential of ORs as therapeutic targets and their key role in mammalian olfaction, there is only a limited number of models designed to predict OR-molecule activity, and no previous approach addresses the challenge of predicting $EC_{50}$. The first studies employ SVM (Kowalewski & Ray, 2020) and random forest (Cong et al., 2022) to predict the responses of limited subsets of ORs and molecules with sufficient *in vitro* data. A subsequent work (Gupta et al., 2021) uses BiLSTM (Graves & Schmidhuber, 2005) to predict the activity of any OR-molecule pair based on SMILES and the receptor's primary structure. The current state-of-the-art (Hladiš et al., 2023) abstracts the OR-molecule interaction as modelling a molecule in a protein-specific environment. It represents the molecular topology as a graph and copies the receptor's [CLS] token from ProtBERT (Elnaggar et al., 2021) to all nodes in the graph. The model then employs a tailored graph neural network to predict the probability that a molecule induces the activity of a receptor. Recently, MAARDTI (Zhan et al., 2025), reported as the best-performing model on drug-target interaction benchmarks, has been applied to predict olfactory receptors' activation. Although competitive, it falls short of the current SOTA, highlighting the challenging nature of OR-molecule activity prediction.

Recently, dose–response modelling has also been investigated for drug–cell inhibitory effects (Alonso Campana et al., 2024). In that setting, the authors assume access to experimental readouts at each concentration and explicitly fit the entire dose–response curve. In contrast, our framework operates in a substantially more restricted regime, where the curve is unknown and only the inflection point of the 4-parameter logistic model (i.e., the $EC_{50}$) or the information that no response is observed is available. Building on this limited supervision, we propose a novel training

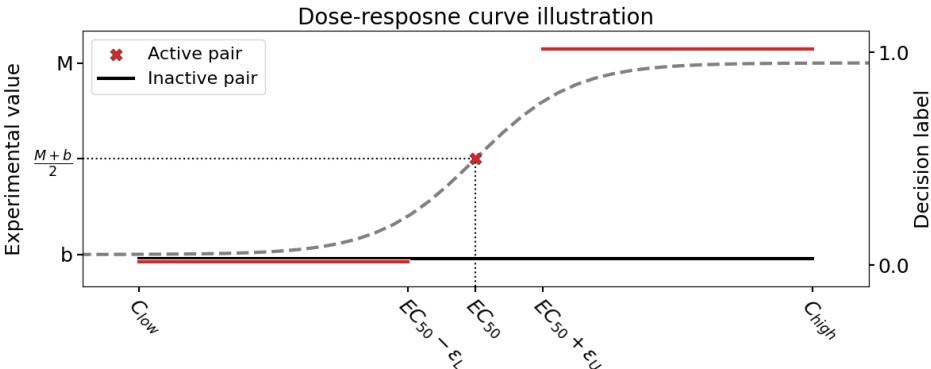

Figure 1: Example of a dose-response curve for an active (grey/red), and inactive (black) pair. The grey line corresponds to the true active curve, which is unknown except for the $EC_{50}$ value (red cross). The red and black lines are concentration intervals corresponding to the training labels on the right y-axis for active and inactive pairs, respectively. The efficacy $M$, basal activity $b$, and half response $\frac{M+b}{2}$ are experimentally measured and not available in the data. In Algorithm 1, a soft label $L \sim Unif(0,1)$ is sampled for concentrations $c \in (EC_{50} - \epsilon_L, EC_{50} + \epsilon_U)$.

and inference strategy that accurately predicts both inflection point and activity decision for a given protein-molecule pair.

## 3 DOSE-RESPONSE CURVE

The relationship between the concentration of a compound and the receptor-mediated response of a cell is typically assessed using functional assays, which are described by dose-response curves. For concentration $c$ on a logarithmic scale, the dose-response curve can be characterised by a generalised logistic model (Neubig et al., 2003):

$$g(c) = \frac{M - b}{1 + 10^{-q(c-EC_{50})}} + b \tag{1}$$

where $M$ is the efficacy, $b$ is the basal activity, $q$ is the slope, and $EC_{50}$ is the 50% effective concentration and the inflection point of the curve. To estimate the parameters of the curve, several concentrations of the compound over several orders of magnitude are tested *in vitro* (Mainland et al., 2014; Saito et al., 2009). However, due to different readouts depending on experimental settings (e.g. luciferase, $Ca^{2+}$, etc.), raw dose-response curve data are rarely available, and most data sources publish only $EC_{50}$ values or information that there is no response and the curve is flat. Formally, we set $EC_{50} = +\infty$ for inactive pairs.

## 4 ESTIMATING $EC_{50}$ AS BINARY DECISION TASK

A widely used approach to train a model to predict $EC_{50}$ is to use $l_2$ loss and estimate the values as a regression. In contrast, we propose an alternative strategy that formulates $EC_{50}$ prediction as a classification task. Similarly to how dose-response curves are obtained from *in vitro* experiments, we model the protein-molecule response at a given molecular concentration, and estimate $EC_{50}$ by fitting (1) to predictions at concentrations spanning several orders of magnitude.

We assume that the model has an access to $EC_{50}$ and the other parameters of the curve 1 are unknown. However, $EC_{50}$ provides a way to draw surrogate decision samples from a normalised dose-response curve $f(c) = \frac{g(c)-b}{M-b}$. The monotonicity of the curve implies that for low concentrations $c \ll EC_{50}$ the protein-molecule pair would be considered inactive *in vitro*, whereas for high concentrations $c \gg EC_{50}$ the pair would be active (Figure 1). Therefore, given a dataset $\mathcal{D} = \{(s^i, m^i), EC_{50}^i\}_i$ of $EC_{50}$ values per each protein-molecule pair $(s^i, m^i)$, we can construct a binary training data $\mathcal{B} = \{(s^i, m^i, c^{i,j}), L^{i,j}\}_{i,j}$ where for each concentration $c^{i,j} \sim Unif(C_{low}, C_{high})$

we sample a decision $L^{i,j} \in \{0,1\}$ whether the pair would be considered active. Then we train a model on $\mathcal{B}$, that predicts probability of activation at a given concentration $P(active|s, m, c)$. Finally, we estimate $EC_{50}$ of a pair $(s^i, m^i)$ by fitting (1) to predictions $\{P(active|s^i, m^i, c^{i,j})\}_j$ at several concentrations, similarly to how the dose-response curve is fitted from *in vitro* experiments. By definition, the probability of activation at $c = EC_{50}$ corresponds to $0.5$, and we uniformly sample $L^{i,j} \sim Unif(0,1)$ for concentrations around $EC_{50}$ based on margins $\epsilon_L$ and $\epsilon_U$. See detailed training and inference procedures in Algorithm 1 and Algorithm 2. Inherently, there is an imbalance between the number of active and inactive pairs and this imbalance is changed in each batch due to the sampling. Thus, we use dynamic sample weights calculated per each batch, with the details given in Section A.2.

The above strategy provides a way to reformulate the traditional regression problem as a classification task that draws samples from a binary surrogate of a normalised dose-response curve. Although $M$ and $b$ used for curve normalisation are generally different for each protein-molecule pair, normalisation $f(c) = \frac{g(c)-b}{M-b}$ only rescales the y-axis, and the inflection point (i.e., $EC_{50}$) remains the same when the curve is fitted during inference in Algorithm 2.

Beyond active pairs with a reported $EC_{50}$, a dose–response assay may show an increased response, but the $EC_{50}$ lies outside the tested concentration range. Therefore, the curve cannot be fitted and only the lower bound $c_t$ on the $EC_{50}$ is available in the data. In such cases, negative samples can still be drawn for $c < c_t$ and we provide further details on this case in Section A.7.

## 5 MODEL

To estimate the concentration-dependent activity of a protein-molecule pair, we consider a model that has 3 inputs: molecular topology $m$, its concentration $c$, and sequence of amino acids $s$, and the output is the probability $P(active|s, m, c)$ that the interaction between the receptor and the molecule at a given concentration $c$ will trigger a response in the cell. We refer to this model as ASMI-DR for *Attention-based Sequence Molecule Interaction for Dose-Response prediction*.

We represent the molecular input as a graph $m = \{\mathcal{V}, \mathcal{E}\}$, where $\mathcal{V}$ is the set of nodes (atoms) and $\mathcal{E}$ the set of edges (bonds). Each node and edge is initialised by feature vectors $x_v$ and $e_{u,v}$, respectively, containing information such as atomic number, bond type, etc. (Table A1). To enhance the expressive power of the network, the graph is oriented and there are two edges $e_{u,v}$ and $e_{v,u}$ between each pair of nodes $u$ and $v$ (Yang et al., 2019). For protein sequence representation, we use ESM-2 (Lin et al., 2023) with frozen weights and the input for a sequence of size $n_s$ is the embedding of the last ESM-2 layer with dimensions $s_{in} \in \mathbb{R}^{n_s \times d}$, where $d$ is the embedding size.

Combining molecule and protein inputs in an early stage of processing turns out to be beneficial for performance (Hladiš et al., 2023). We build upon this observation in the architecture outlined in Figure 2. The molecular graph is first transformed by the GNN embedding block (Figure A1) and, together with the sequence embedding, they are processed via a series of cross blocks. Each cross block is composed of a node update block (Figure A2a) and a sequence update block (Figure A2b). The core of these blocks is multi-head cross-attention, which learns changes in the molecular node and amino acid embeddings induced by the interactions with the sequence and molecule, respectively. In the node update block, node embeddings are used as queries in cross-attention, and amino acid embeddings are keys and values. The signal is then passed through the residual connection and feed forward network (FFN) and finally the updated node representation is processed by a graph isomorphism network (GIN) (Xu et al., 2019), which allows exploiting the graph structure and edge information that are not considered in cross-attention. The sequence update block is designed analogously. The sequence embedding is transformed in the cross-attention layer, where amino acids are queries, and molecular nodes are keys and values. The updated sequence representation is then processed by a self-attention layer (Vaswani et al., 2017) which can be interpreted as a graph neural network applied to a fully-connected graph where all amino acids are linked to each other.

An additional input to the cross blocks is concentration $c$, which is used in the form of auxiliary query features in cross-attention. It is first mapped to a vector $cw \in \mathbb{R}^{d_c}$ where $w$ is learned, but its norm depends on the input concentration. Then $cw$ is concatenated to each query just before multi-head cross-attention (green rectangle in Figure A2), allowing the model to learn the concentration-dependent protein-molecule interaction.

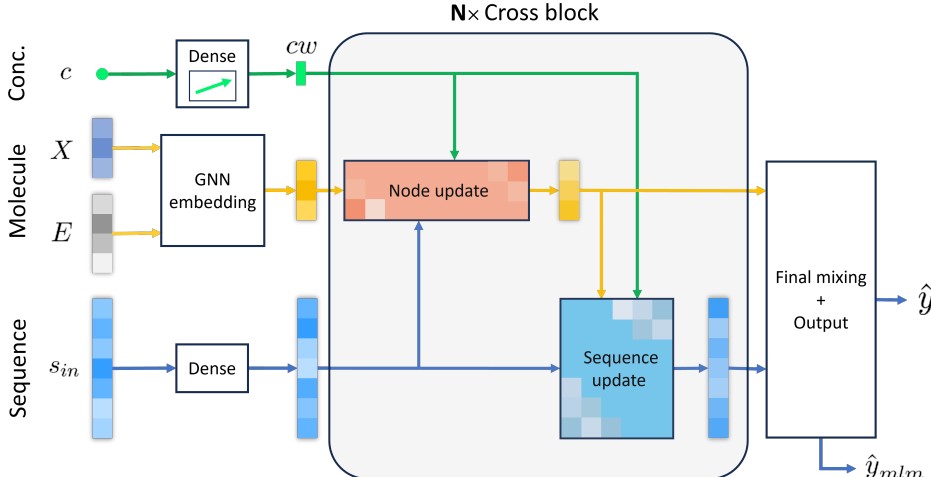

Figure 2: Outline of ASMI-DR architecture. The inputs to the model are molecular graph $m$ with node features $X$ and edge features $E$, molecular concentration $c$, and sequence embedding $s_{in}$. The molecular graph is passed through the GNN embedding layer and then processed together with the sequence representation and concentration in a series of cross blocks. In each cross block, the concentration-dependent interaction between the molecule and the protein is modelled by iteratively updating node and sequence embeddings in update blocks, described in Figure A2. The updated node and sequence representations are then concatenated and passed through multi-head attention and finally two output heads for response prediction and masked language modelling (MLM).

Finally, after $N$ cross blocks, the node and sequence representations are combined in the mixing block followed by the output heads (Figure A3). Here, node and amino acid embeddings are concatenated together and passed to multi-head self-attention, followed by residual connection and FFN. In cross-attention, the softmax is either performed per nodes or per amino acids and the model is restricted to learning the interaction between the molecule and the sequence. In contrast, softmax in the mixing block is performed through nodes and amino acids simultaneously and could give weight to self-interactions within the molecule and sequence (yellow and purple rectangles in Figure A3).

The final concatenated embedding from the mixing block is passed to a pooling layer[1] and MLP to obtain the final prediction. In addition, the sequence representation is extracted from the output of the mixing block and used for the masked language modelling (MLM) task (Figure A3) (Devlin et al., 2019). MLM has been shown to extract protein structure information from the sequence of amino acids (Vig et al., 2021; Lin et al., 2023) and to avoid forgetting this information during training, we add the MLM task as an auxiliary training goal. We employ a common strategy to replace an amino acid by a [MASK] token with 15% chance and then predict the probability $\hat{y}_{mlm}^i$ that the given amino acid was at the position $i$ in the original sequence. Apart from retaining the structural information, masking amino acids also serves as a training perturbation to avoid overfitting.

## 6    DATASET

We train and evaluate the proposed algorithm on M2OR version v1.2.0 (Lalis et al., 2024a), which is a curated dataset gathering functional assay data on 77611 experiments for 1402 protein sequences and 771 molecules. After preprocessing described in Section A.4, M2OR contains 5773 $EC_{50}$ samples out of which 1427 are active and 4346 correspond to inactive pairs. In addition, preprocessed M2OR lists 60256 primary and secondary screening experiments, which are further discussed in Section A.7. To further validate our framework, we evaluate ASMI-DR on drug-target affinity (DTA) datasets: DAVIS Davis et al. (2011) and BindingDB ($K_d$) Gilson et al. (2015); Huang et al. (2021), which after preprocessing gather 25772 and 42234 experimentally measured dissociation constants

---

[1]We use attention pooling from Eqn. (2) in (Hladiš et al., 2023).

($K_d$), respectively. Furthermore, we evaluate a concentration-free version of the proposed model on a drug-target interaction (DTI) benchmark, KIBA (Tang et al., 2014).

# 7 EXPERIMENTS

We perform a series of experiments to validate our training and inference algorithms and the proposed model architecture. We consider two evaluation tasks: activity decision prediction and $EC_{50}$ (or $K_d$) estimation. The latter, presented in Section 7.2 and Section 7.3, is assessed by calculating the root mean squared log error (RMSLE) between the predictions and the experimentally measured values for the active pairs in the test set. The former, discussed in Section 7.1, evaluates whether the decision about the activity based on Algorithm 2 aligns with the experimental activity decision. Finally, in Section A.7 we consider extending our pipeline with screening data and, in Section A.6, we test a version of the ASMI-DR architecture without concentration input on DTI benchmarks.

The test sets in all experiments contain approximately 20% of the data and they comprise only dose-response assays, excluding any screening campaigns. We perform 5 cross-validation runs for each experiment. We use 2 cross blocks, sequences are padded to $n_s = 512$ and the sequence embedding is initialised by ESM-2 with 33 layers and $d = 1280$.

**Dose-response curve fit.** To run the inference in Algorithm 2, we predict the responses of the protein-molecule pairs at molecular concentrations spanning range $10^{-8}$M to $10^{-2}$M for M2OR and $10^{-12}$M to $10^{-4}$M for DAVIS and BindingDB. This covers 98.5% of all available $EC_{50}$ values in M2OR, and 100% and 99.9% of $K_d$ values in DAVIS and BindingDB, respectively. We exclude values outside of these ranges from test sets.

**Baselines.** For the activity decision evaluation, we compare our approach with several DTI models and previous works that predict OR-molecule activation without taking into account concentration: HyperAttentionDTI (Zhao et al., 2021), MolTrans (Huang et al., 2020), MAARDTI (Zhan et al., 2025), BiLSTM (Gupta et al., 2021), GNN-CLS (Hladiš et al., 2023). We also report the performance of a version of our architecture, where we omit the concentration input to the model (ASMI-Prob). For M2OR, we train this model on the pair table which is available as a part of the dataset (Lalis et al., 2024a). To consider receptors' tertiary structure, we evaluate the performance of the probability of binding from the state-of-the-art cofolding model Boltz-2 (Passaro et al., 2025) and the decision based on docking using SMINA with Vinardo scoring function (Masters et al., 2020; Quiroga & Villarreal, 2016) on AlphaFold2 (Jumper et al., 2021) structures (see details of the protocol in Section A.5.2). We perform docking on 868 (300 active and 568 inactive) pairs corresponding to the OR sequences available at AlphaFold DB (Varadi et al., 2023; 2021). We generated up to 9 poses and use the Youden index estimated on 10% of the data to transform docking scores into activity decisions. Finally, we compare the results to the widely used experimental protocol, where we consider *in vitro* screening runs as a predictor for the dose-response activity decision.

As a baseline for the $EC_{50}$ and $K_d$ estimation, we compare our approach to state-of-the-art DTA models: DeepDTAGen (Shah et al., 2025), DTIAM (Lu et al., 2025), and ProSmith (Kroll et al., 2024). Furthermore, to rule out the influence of the architecture on the evaluation of our framework, we also report the performance of the concentration-free version of ASMI architecture trained in a regression task (ASMI-Reg). During training of regression baselines on M2OR, we set the $EC_{50}$ of inactive pairs to 1M, and we also report the performance of this model in the activity decision task by considering pairs with the estimated $EC_{50} > 10^{-1}$M as inactive. Similarly for DAVIS and BindingDB, we follow standard practices (Huang et al., 2021) and we set $K_d$ of inactive pairs, i.e., the pairs where no interaction was observed in the tested concentration range, to 10 $\mu$M. We consider pairs with the estimated $K_d > 7.94\mu$M as inactive[2].

## 7.1 ACTIVITY DECISION

A major goal of a dose-response experiment is to assess whether a molecule "activates" a given protein of interest, i.e., whether the interaction between the protein and the molecule at any high-enough concentration elicits a response in the cell. We assess the capability of our approach in

---

[2]$10\mu$M corresponds to 1 log($\mu$M) unit and $7.94\mu$M to 0.9 log($\mu$M) units.

Table 1: Evaluation of the activity decision task. *Primary sc.* and *Secondary sc.* stand for primary and secondary *in vitro* screening, respectively. *Naive* is the performance using the frequency statistics, i.e., $P(active|s, m) = \hat{p}_s \hat{p}_m$, where $\hat{p}_s$ and $\hat{p}_m$ are marginal probabilities for a given protein and molecule, respectively (Lalis et al., 2024b). The results for MAARDTI are taken from the original publication (Zhan et al., 2025) and the results for BiLSTM, MolTrans, HyperAttentionDTI, and CLS-GNN are taken from (Hladiš et al., 2023). Standard deviation is given in parentheses.

| Model | Precision | Recall | F-score | AveP | MCC |
|---|---|---|---|---|---|
| Primary sc. | 0.56 | 0.40 | 0.47 | - | 0.24 |
| Secondary sc. | 0.70 | 0.45 | 0.55 | - | 0.48 |
| Docking[a] | 0.45 (0.02) | 0.65 (0.11) | 0.53 (0.03) | 0.42 (0.01) | 0.22 (0.02) |
| Boltz-2 | 0.54 (0.12) | 0.05 (0.01) | 0.09 (0.01) | 0.38 (0.03) | 0.11 (0.03) |
| Naive | 0.27 (0.03) | 0.60 (0.05) | 0.37 (0.02) | 0.22 (0.02) | 0.25 (0.03) |
| BiLSTM | 0.34 | 0.71 | 0.46 | - | - |
| MolTrans | 0.40 (0.05) | 0.82 (0.03) | 0.56 (0.05) | 0.64 (0.07) | 0.48 (0.04) |
| MAARDTI | 0.70 | 0.60 | 0.64 | 0.70 | 0.56 |
| HyperAttentionDTI | 0.61 (0.03) | 0.77 (0.02) | 0.68 (0.02) | 0.74 (0.02) | 0.58 (0.02) |
| GNN-CLS | 0.69 (0.02) | 0.70 (0.04) | 0.69 (0.02) | 0.78 (0.01) | 0.61 (0.02) |
| ASMI-Prob | 0.72 (0.07) | 0.73 (0.04) | 0.72 (0.02) | **0.80** (0.04) | 0.63 (0.04) |
| ASMI-Reg | 0.69 (0.05) | 0.72 (0.04) | 0.71 (0.04) | 0.70 (0.01) | 0.62 (0.05) |
| ASMI-DR | 0.77 (0.03) | 0.72 (0.05) | **0.75** (0.02) | 0.75 (0.02) | **0.67** (0.02) |

[a]Performance on 781 pairs on average corresponding to the OR sequences available at AlphaFold DB.

this decision task by using the fitted curve's maximum $\hat{M}$ from Algorithm 2. We consider a given protein-molecule pair to be "active" if $\hat{M} > 0.5$ and "inactive" otherwise. If the curve cannot be fitted, we consider the pair to be inactive. Due to the imbalanced activity label distribution, we use Matthews Correlation Coefficient (MCC) as the main metric in the activity decision experiments.

**Comparison with the state-of-the-art.** We benchmark ASMI-DR trained using our pipeline against existing models for predicting OR-molecule activity, with results summarised in Table 1. Our approach demonstrates superior performance, achieving a $10\%$ improvement in MCC over the state-of-the-art GNN-CLS. While GNN-CLS and ASMI-Prob show higher Average Precision (AveP), they lag behind ASMI-DR in other key metrics, including F-score and MCC. Furthermore, our approach outperforms its regression variant, ASMI-Reg, by $8\%$ in MCC.

**Novel protein sequence generalisation.** To assess the ability of our pipeline to generalise to new protein sequences, we adopt the out-of-distribution (OOD) evaluation procedure from Hladiš et al. (2023), and report the results in two scenarios. The first scenario, Sequence–Single in Table 2, evaluates intra-family generalisation. In this setup, randomly chosen protein sequences are held out exclusively for testing, ensuring the model is evaluated on unseen but related proteins. This mimics a realistic scenario where the model must predict responses for new members of a protein family based on sequence similarity. The second and more challenging scenario, Sequence–Cluster in Table 2, tests inter-family generalisation. Here, we cluster proteins by sequence similarity and hold out 5 entire clusters for testing. This forces the model to extrapolate to protein families that are dissimilar to any sequence seen during training.

As can be seen in Table 2, ASMI-DR outperforms previous approaches in the Single scenario, surpassing the second-best GNN-CLS model by $15\%$. In the demanding Cluster scenario, ASMI-DR ranks second after its regression variant, and it surpasses the current state-of-the-art.

**Novel compound generalisation.** Analogously to the generalisation to new protein sequences, we evaluate the ability of our approach to accurately predict the responses of previously unseen compounds. The first scenario, Molecule–Single in Table 2, simulates the generalisation to small changes in the molecular structure, a situation of particular interest in olfaction where small struc-

Table 2: $EC_{50}$ estimation and out-of-distribution evaluation. *Mean model* stands for a naive baseline that assigns the mean $EC_{50}$ to all protein-molecule pairs. *$EC_{50}$ error* is the experimental error and the performance lower bound. The results for MAARDTI and GNN-CLS are taken from the respective publications. Values in parentheses represent standard deviation.

| Datacase | | Name | MCC ↑ | Precision ↑ | RMSLE ↓ | Spearman's $\rho$ ↑ |
|---|---|---|---|---|---|---|
| *in vitro* | | Primary sc. | 0.238 | 0.563 | | |
| | | Secondary sc. | 0.476 | 0.704 | | |
| | | $EC_{50}$ error | | | 0.334 | |
| i.i.d. | | Mean model | | | 0.899 (0.025) | |
| | | Boltz-2 | 0.108 (0.033) | 0.541 (0.117) | $1.110^{a}$ (0.037) | $0.148^{a}$ (0.052) |
| | | MAARDTI | 0.555 | 0.700 | | |
| | | GNN-CLS | 0.605 (0.02) | 0.689 (0.02) | | |
| | | DTIAM | 0.658 (0.029) | 0.646 (0.053) | 1.516 (0.093) | 0.363 (0.022) |
| | | DeepDTAGen | 0.592 (0.043) | 0.631 (0.079) | 1.417 (0.243) | 0.224 (0.065) |
| | | ProSmith | 0.654 (0.030) | 0.649 (0.041) | 1.402 (0.111) | 0.355 (0.045) |
| | | ASMI-Reg | 0.621 (0.048) | 0.692 (0.047) | 1.213 (0.135) | 0.399 (0.122) |
| | | ASMI-DR | **0.671** (0.016) | 0.773 (0.028) | **0.725** (0.070) | **0.648** (0.065) |
| Sequence | Single | MAARDTI | 0.323 | 0.757 | | |
| | | GNN-CLS | 0.417 (0.01) | 0.636 (0.07) | | |
| | | ASMI-Reg | 0.398 (0.112) | 0.506 (0.142) | 1.543 (0.512) | 0.150 (0.281) |
| | | ASMI-DR | **0.481** (0.031) | 0.642 (0.038) | **0.761** (0.150) | **0.470** (0.119) |
| | Cluster | MAARDTI | 0.147 | 0.545 | | |
| | | GNN-CLS | 0.088 (0.06) | 0.535 (0.12) | | |
| | | ASMI-Reg | **0.238** (0.123) | 0.362 (0.113) | 1.889 (0.481) | -0.145 (0.158) |
| | | ASMI-DR | 0.218 (0.043) | 0.461 (0.162) | **1.170** (0.269) | **0.040** (0.150) |
| Molecule | Single | MAARDTI | 0.409 | 0.633 | | |
| | | GNN-CLS | 0.533 (0.07) | 0.657 (0.11) | | |
| | | ASMI-Reg | 0.531 (0.054) | 0.572 (0.086) | 1.729 (0.347) | 0.286 (0.152) |
| | | ASMI-DR | **0.593** (0.074) | 0.663 (0.098) | **0.920** (0.096) | **0.474** (0.116) |
| | Cluster | MAARDTI | **0.399** | 0.795 | | |
| | | GNN-CLS | 0.334 (0.07) | 0.544 (0.07) | | |
| | | ASMI-Reg | 0.395 (0.082) | 0.548 (0.151) | 1.561 (0.371) | 0.154 (0.107) |
| | | ASMI-DR | 0.398 (0.077) | 0.572 (0.115) | **0.818** (0.154) | **0.298** (0.116) |

[a]Due to the low MCC, the evaluation is also done on incorrectly classified pairs.

tural changes can lead to a significant difference in odour perception (Sell, 2006). It is constructed by randomly selecting molecules and placing all their occurrences in the test set. The second, and more challenging scenario, Molecule–Cluster in Table 2, tests generalisation to new chemical scaffolds. For this, we cluster compounds by structural similarity using the Tanimoto coefficient and hold out 5 clusters for testing, forcing the model to extrapolate to unseen scaffolds.

According to the results presented in Table 2, ASMI-DR is on par with the previous approaches in the Cluster scenario, and it outperforms the best baseline by 11% in the Single scenario. Notably, it scores only 2% lower in MCC in the Single scenario compared to the current state-of-the-art evaluated on the less challenging i.i.d. case.

**Comparison to structure models.** Table 1 reveals that docking and Boltz-2 show poor performance in predicting activity, and ASMI-DR outperforms them by a large margin. Although, the advantage of these approaches is that they are zero-shot, we observe that even for OOD evaluation, ASMI-DR achieves better performance in terms of MCC in all OOD scenarios except Sequence - Cluster, where it is on par with docking and still outperforms Boltz-2. A possible reason for the low performance of structure-based models is, that both approaches are designed to predict binding

events, and in GPCRs, binding of a molecule to the protein is a necessary but not sufficient condition for the activation (Zhang et al., 2024).

**Comparison to *in vitro* screening.** A standard procedure in the wet lab is to perform primary and secondary screening rounds to select candidates for dose-response experiments (additional details in Section A.7). In the context of activity decision, one can view screening as a predictor and ask the question: what is the performance of our approach compared to the screening rounds? Notably, ASMI-DR outperforms costly *in vitro* primary and secondary screening rounds in an i.i.d. evaluation (Table 1). Even in OOD settings (Table 2), ASMI-DR outperforms the screening rounds in cases where novel but structurally similar molecules/proteins to the training ones are tested (Molecule–Single and Sequence–Single). In case new molecular scaffolds are tested on previously explored proteins (Molecule–Cluster), ASMI-DR falls behind secondary screening, but outperforms primary screening by a large margin of 67% in MCC.

## 7.2 $EC_{50}$ ESTIMATION

The second evaluation criterion is the pipeline's ability to accurately predict $EC_{50}$. We compare the performance of the ASMI-DR trained using Algorithm 1 to state-of-the-art DTA models DTIAM, DeepDTAGen and ProSmith, to ASMI-Reg trained in a regression task, and to a naive baseline, which assigns a mean of $EC_{50}$ values in the training set to all test set pairs. We also report the *in vitro* $EC_{50}$ measurement error as a lower bound. The reported performance is evaluated on protein-molecule pairs that the models correctly identified as active, since $EC_{50}$ for inactive pairs cannot be experimentally measured and it is replaced by an arbitrary value of 1M in the regression training.

The results of the $EC_{50}$ evaluation in Table 2 reveal that ASMI-DR outperforms its regression counterpart ASMI-Reg by 40% in RMSLE, and it surpasses all the evaluated DTA baselines by half an error. Affinity module of Boltz-2 also outperforms the regression model, but the error of ASMI-DR remains 0.385 log units below Boltz-2. A similar observation holds even for the OOD evaluation, where our pipeline consistently outperforms its regression variant by a margin and achieves a lower error than Boltz-2 in all OOD scenarios, except for the most challenging Sequence–Cluster scenario. The same conclusion can be drawn for Spearman's rank correlation, which is of a particular interest in olfaction, where receptors with the lowest $EC_{50}$ encode odour identity (Wilson et al., 2017; Zwicker, 2019). To further analyse the rank correlation, we evaluate the mean Spearman's $\rho$ per sequence/molecule in Table A5 in Section A.8.

Hladiš et al. (2023) explore the possibility to use screening data in the activity decision task. Algorithm 1 treats each triplet $(s, m, c)$ independently, and as such, it allows for a principled integration of screening data in the model training. In Section A.7, we extend Algorithm 1 to include screening data and we evaluate it in Section A.7.1. Although screening constitutes the majority of the M2OR dataset, its integration does not improve the performance in the activity decision task and it marginally lowers the error in the $EC_{50}$ estimation task (Table A4). A possible explanation is that for screening at concentration $c_s$, the labels are sampled "one-sided" in a subset of the sampling region $[C_{low}, c_s]$ or $[c_s, C_{high}]$. This leads to a severe label imbalance, especially when $c_s$ is close to the sampling region boundaries.

## 7.3 $K_D$ ESTIMATION

To further challenge the proposed framework, we compare the performance of ASMI-DR with state-of-the-art DTA models on DAVIS and BindingDB datasets. In analogy with $EC_{50}$ from GPCR functional assays, $K_d$ in log-units obtained via competitive binding assays can be modelled by the generalized logistic curve 1 with decreasing slope $q < 0$ (Fabian et al., 2005). Thus, we can train ASMI-DR by sampling label 1 for concentrations greater than $K_d$ in Algorithm 1 and modelling $P(binding|s, m, c)$ instead of $P(active|s, m, c)$.

As shown in Table 3, ASMI-DR achieves $K_d$ estimation RMSLE that is on par with the best-performing regression baseline ProSmith on both DAVIS and BindingDB (paired $t$-test $p$-values of 0.106 and 0.315, respectively), while consistently outperforming all competing methods on the activity decision metrics (MCC and precision). On BindingDB in particular, ProSmith exhibits a tendency to assign overly low $K_d$ values even to inactive pairs, which improves its RMSLE but leads to reduced MCC. In contrast, ASMI-DR produces accurate activity predictions, achieving the high-

Table 3: $K_d$ estimation evaluation. Standard deviation is given in parentheses.

| Dataset | Model | MCC ↑ | Precision ↑ | RMSLE ↓ | Spearman's $\rho$ ↑ |
|---|---|---|---|---|---|
| DAVIS | DTIAM | 0.566 (0.014) | 0.543 (0.013) | 0.736 (0.010) | 0.672 (0.017) |
| | DeepDTAGen | 0.642 (0.014) | 0.691 (0.015) | 0.788 (0.017) | 0.617 (0.027) |
| | ProSmith | 0.631 (0.010) | 0.619 (0.013) | **0.688** (0.004) | **0.700** (0.012) |
| | ASMI-Reg | 0.601 (0.048) | 0.646 (0.063) | 0.772 (0.057) | 0.639 (0.023) |
| | ASMI-DR | **0.685** (0.005) | 0.772 (0.016) | 0.713 (0.028) | 0.696 (0.014) |
| BindingDB | DTIAM | 0.606 (0.026) | 0.678 (0.021) | 0.820 (0.020) | 0.779 (0.009) |
| | DeepDTAGen | 0.722 (0.014) | 0.827 (0.018) | 0.865 (0.029) | 0.748 (0.009) |
| | ProSmith | 0.607 (0.041) | 0.678 (0.033) | **0.808** (0.026) | **0.785** (0.013) |
| | ASMI-Reg | 0.446 (0.231) | 0.620 (0.140) | 0.926 (0.116) | 0.699 (0.081) |
| | ASMI-DR | **0.745** (0.016) | 0.839 (0.018) | 0.834 (0.032) | 0.774 (0.013) |

est MCC on both datasets with only a marginal difference in RMSLE and Spearman's $\rho$ relative to ProSmith.

Relative to its regression counterpart ASMI-Reg, ASMI-DR yields sizeable gains across all metrics, highlighting the benefit of the proposed framework. Moreover, the MCC of ASMI-DR on DAVIS also surpasses that of dedicated classification DTI models (Table A2), indicating that incorporating the dose-response training objective not only preserves affinity estimation accuracy but also leads to superior binary activity prediction.

## 8 CONCLUSION

In this study, we reformulate an $EC_{50}$ regression task as a series of binary classifications that mimic *in vitro* dose-response assays. Leveraging the monotonicity of the dose-response curve, we first sample binary activity labels at a given molecular concentration, and then train a novel model architecture that estimates the probability $P(active|s, m, c)$ that a molecule activates a protein at a concentration $c$. Querying this model at concentrations spanning several orders of magnitude enables fitting a generalised logistic model 1, which yields both the final predicted activity (curve's maximum) and $EC_{50}$ (curve's inflection point) in a unified framework.

We demonstrate that training a model according to our proposed Algorithm 1 simultaneously outperforms both state-of-the-art activity prediction models and $EC_{50}$ regression baselines on the challenging M2OR dataset. The proposed pipeline even surpasses a common *in vitro* primary screening in out-of-distribution settings, achieving errors below one log unit.

Algorithm 1 provides a principled way how to include screening outcomes in the model training. However, a naive extension of the Algorithm 1 to screening data introduces a substantial label imbalance at concentrations near the boundaries of the sampling interval, and a future work can refine the sampling procedure for abundant screening data. While our predictions align well with experimental measurements, the current pipeline does not yet account for receptor inhibition or multi-ligand interactions (e.g., allosteric modulation, competitive antagonism). It also introduces computational overhead at inference time, as it requires multiple queries across different concentrations to estimate a single $EC_{50}$ value. However, the computational costs are substantially lower compared to structure-based approaches such as Boltz-2, which ASMI-DR surpasses by 0.385 in RMSLE.

Olfactory receptors (ORs) form the largest subfamily of GPCRs – targets for nearly 30% of approved drugs – yet 48% of tested OR sequences remain orphaned without a known ligand (Lalis et al., 2024b). In olfaction, revealing the response of these proteins is crucial in deciphering the odour coding and solving the ever-present challenge of smell prediction. Our approach opens a new avenue for characterising both odour coding and the drugability of ORs.

## REPRODUCIBILITY STATEMENT

The anonymised code is attached with the submission or available here: `https://drive.google.com/file/d/11d7RkYYG-vlY2OYnMPYhiLgaHKGpAdx4/view?usp=drive_link`. Preprocessed data and all splits used in the work can be downloaded from the links in the readme. Data preprocessing steps are described in Section A.4. All baseline models were either run with the default hyperparameters or the results were taken from the corresponding publication, as indicated in table captions.

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

# A APPENDIX

## A.1 SAMPLING PSEUDO-CODE

---

**Algorithm 1:** Training

---

**Input:** Dataset $\mathcal{D} = \{(s^i, m^i), A^i, l^i\}_i$ with molecules $m^i$, sequences $s^i$, $EC_{50}$ value $A^i$, and activity label $l^i$
**Input:** Concentration range $(L, U)$
**Input:** Margins $\varepsilon_L, \varepsilon_U$
$X = [], \quad Y = []$
**for** $batch \sim \mathcal{D}$ **do**
  **for** $i$, $(m, s)$, $A$, $label$, **in** $enumerate(batch)$ **do**
    /* Set negative examples
       and truncate to $(L, U)$
    **if** $label = 0$ **then** $A \leftarrow U + \varepsilon_L$
    **if** $A > U + \varepsilon_L$ **then** $A \leftarrow U + \varepsilon_L$
    **if** $A < L - \varepsilon_U$ **then** $A \leftarrow L - \varepsilon_U$
    /* Sample a concentration $c$
       and a training label $l_{tr}$
    $c \sim Unif(L, U)$
    **if** $c \leq A - \varepsilon_L$ **then** $l_{tr} \leftarrow 0$
    **else if** $c \geq A + \varepsilon_U$ **then** $l_{tr} \leftarrow 1$
    **else** $l_{tr} \sim Unif(0, 1)$
    $X[i] \leftarrow (m, s, c), \quad Y[i] \leftarrow l_{tr}$
  **end**
  /* Get batch sample weights
    based on probability of
    drawing a label for a
    given concentration.
  $W \leftarrow Get\_sample\_weights(batch)$
  /* Train as a standard
    classification task
  $\hat{Y} \leftarrow model(X)$
  $loss \leftarrow loss\_func(\hat{Y}, Y, W)$
  /* Take gradient of the loss
    and update model weights.
**end**

---

**Algorithm 2:** Inference

---

**Input:** Moleclue-sequence pair $(m, s)$
**Input:** Set $C$ of concentrations uniformly covering range $(L, U)$
**Function** DS_curve ($c$, $A$, $q$, $M$, $b$):
  **return**
  $(M - b)/(1 + pow(10, -q(c - A))) + b$
$Y = []$
**for** $i$, $c$ **in** $enumerate(C)$ **do**
  $Y[i] \leftarrow model((m, s, c))$
**end**
/* Use any standard algorithm to
  fit a curve
$\hat{A}, \hat{q}, \hat{M}, \hat{b} \leftarrow fit(\text{DS\_curve}, x = C, y = Y)$
/* Return the estimated $EC_{50}$ and
  the probability of activity
**return** $\hat{A}, \hat{M}$

---

## A.2 SAMPLE WEIGHTS

Prediction of the protein-molecule response is inherently an imbalanced problem, and most of the experimental data are inactive. To account for the label distribution, we use standard imbalance ratio weights (Fernández et al., 2018). However, the sampling presented in Algorithm 1 dynamically generates labels in each iteration and changes the raw data label distribution, since active $EC_{50}$ pairs can contribute to the inactive labels. Therefore, we calculate class weights on the fly for each batch by estimating the probability $P(l|c)$ that a label $l$ is sampled for a given concentration $c$.

Formally, denote the truncated experimental $EC_{50}$ values as $A^i_{trunc} = trunc(EC^i_{50}, (C_{low} - \epsilon_U, C_{high} + \epsilon_L))$. Then the probability that a label $l$ is sampled for a concentration c in a batch

of size $n$ is given by a number of times $c$ is below or above the truncated $EC_{50}$:

$$P(l = 0|c) \approx \frac{\#\{i|c \leq A^i_{trunc} - \epsilon_L\}}{n} \tag{2}$$

$$P(l = 1|c) \approx \frac{\#\{i|c \geq A^i_{trunc} + \epsilon_U\}}{n} \tag{3}$$

$$P(l \in (0, 1)|c) = 1 - P(l = 0|c) - P(l = 1|c) \tag{4}$$

Note that $EC_{50}$ of inactive pairs is set to $+\infty$, thus $A^i_{trunc} = U + \epsilon_L$ for these pairs. Given the above estimate of the label probability, we set the class weights for each triplet $(s, m, c)$ as

$$w_{class}(l, c) = \frac{1}{2P(l|c)} \tag{5}$$

### A.3 MODEL DETAILS

#### A.3.1 MODEL ARCHITECTURE

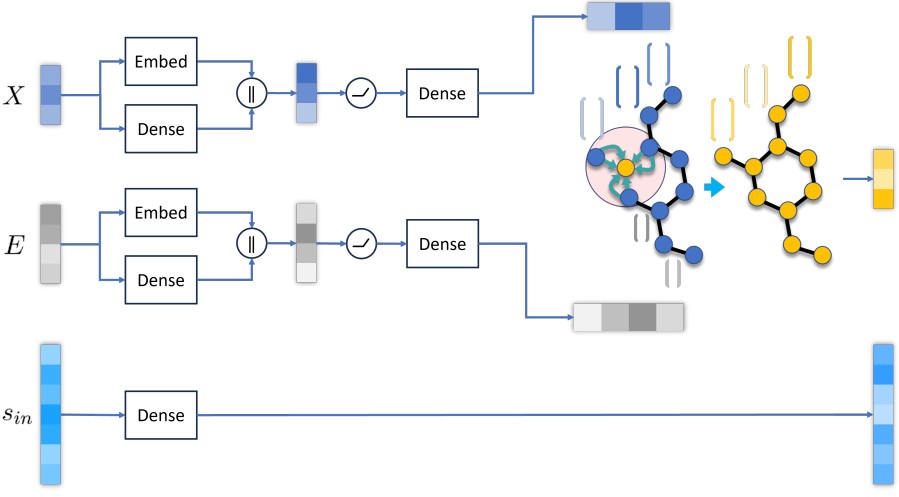

Figure A1: GNN embedding block. Node and edge feature matrices $X$ and $E$, respectively, are embedded and processed by a message passing neural network (MPNN) (Gilmer et al., 2017). Categorical features are passed through a standard embedding layer and continuous features are passed through a dense layer before being concatenated together. The sequence representation, comprising a $d$-dimensional vector per each amino acid, is passed through a single dense layer to form the input to the subsequent cross block. MPNN is composed of an edge update function that concatenates edge features, incoming node features, and outgoing node features and passes them through a dense layer followed by ReLU. It uses GRU (Cho et al., 2014) as a node update function. Symbols ‖, ⊘ stand for row-wise concatenation and ReLU activation function, respectively.

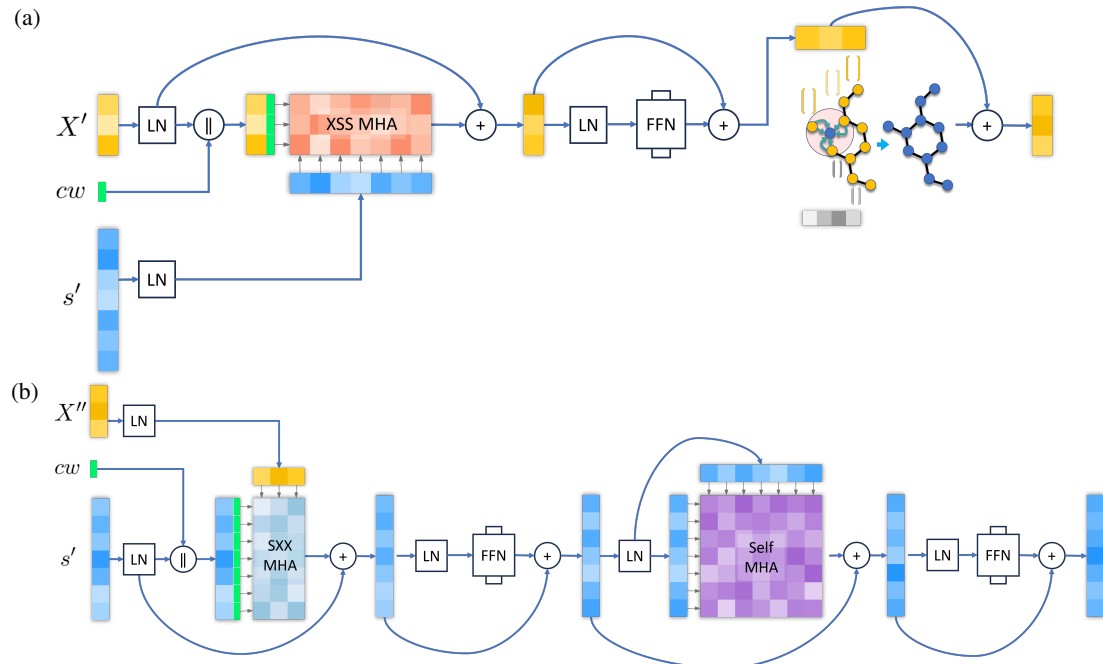

Figure A2: Update blocks. (a) Node update block. The inputs are the node embedding matrix $X'$, the sequence representation matrix $s'$, and the concentration vector $cw$. $X'$ is passed through Layer norm (LN) (Ba et al., 2016) and concatenated with $cw$ to become queries in multi-head cross-attention (XSS MHA) (Vaswani et al., 2017). Sequence representation $s'$ serves as keys/values, and the output of cross-attention is added to the normalised node embeddings via a residual connection (He et al., 2015). The purpose of XSS MHA is to learn the difference in the node representation that is induced by the interaction with the protein. The resulting updated node embeddings are then processed by feed-forward network (FFN) (Vaswani et al., 2017) and together with edge embeddings $E$ form input to Graph isomorphism network (GIN) (Xu et al., 2019). (b) Sequence update block. Analogously, the inputs to the block are the node embedding matrix $X''$, the sequence representation matrix $s'$, and the concentration vector $cw$. $s'$ is passed through Layer norm (LN) and concatenated with $cw$ to become queries in multi-head cross-attention (SXX MHA). Node embeddings $X''$ are keys/values in cross-attention, and the output of SXX MHA is added to the normalised sequence representation. The goal of SXX MHA is to learn the difference in the sequence representation induced by the interaction with the molecule at the given concentration. The updated sequence embedding is then passed through FFN and then to a Transformer block consisting of self-attention (Vaswani et al., 2017) and a second FFN.

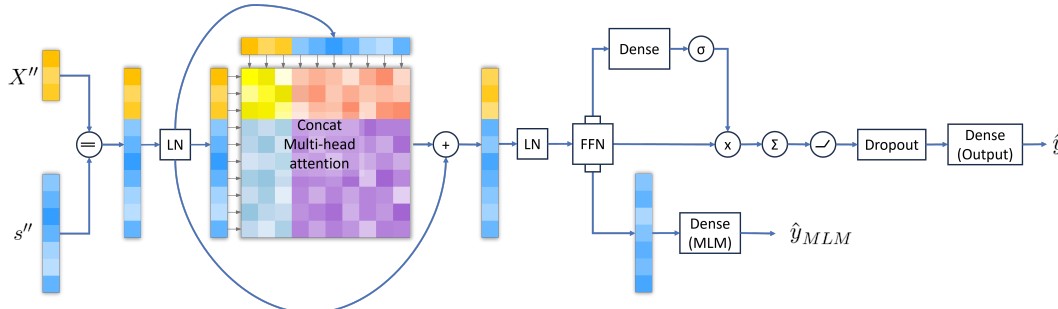

Figure A3: Final mixing block. The updated node embedding matrix $X''$ and the updated sequence representation $s''$ are concatenated together and form an input to self-attention (Vaswani et al., 2017). Note that unlike the case of update blocks, softmax in this self-attention is performed simultaneously through both molecular node and amino acid embeddings. The output of the attention layer is passed through FFN, aggregated via attention pooling (Eqn. (2) in (Hladiš et al., 2023)), and finally processed by a dense layer for a final activity prediction. The part of the FFN output corresponding to the sequence representation is also used for masked language modelling task (MLM). The symbols $\ominus, \oslash, \circledcirc, \otimes, \Sigma$ stand for column-wise concatenation, ReLU activation function, sigmoid activation function, elementwise multiplication and summation, respectively.

### A.3.2 MOLECULAR FEATURES

Table A1: Initial node and edge features. *Cat.* stands for categorical features.

|  | **Atom features** | **Bond features** |
|---|---|---|
| Cat. | Atomic number
Chiral tag
Hybridisation
Is aromatic | Bond type
Stereo type
Is aromatic |
| Continous | Formal charge
Num. of implicit Hs
Explicit valence
Mass | |

### A.3.3 HYPERPARAMETERS AND ASMI ARCHITECTURAL CHOICES

During the ASMI architecture development we experimented with several choices, such as MPNN (Gilmer et al., 2017), GIN (Xu et al., 2019) or GAT (Veličković et al., 2018) as the choice of the GNN layer in Node update block, and PNA (Corso et al., 2020) and attention pooling (Hladiš et al., 2023) as the final pooling function. A systematic sweep over sampling hyperparameters $\epsilon_L$ and $\epsilon_U$ was not performed.

### A.3.4 ASMI-DR TRAINING

ASMI-DR is trained according to Algorithm 1 with $1500$ epochs, batch size of $1024$, and margins $\epsilon_L = \epsilon_U = 0.25$. During training, we apply dropout of $0.1$ in the attention layers, $0.2$ in the FFN layers, and $0.5$ just before the output layer. Padding is set to 32 nodes and 64 edges, which covers all molecules in the M2OR dataset. The initial learning rate is set to $\frac{1}{\sqrt{256}}$ and we use 6000 warm-up steps for the scheduler. The best epoch was chosen based on the validation set comprising $10\%$ of randomly selected pairs. All models are implemented in JAX (Bradbury et al., 2018) and FLAX (Heek et al., 2024) and they were trained on Nvidia A100 SXM4 80GB GPUs or Nvidia H100 NVL 94GB GPUs.

### A.3.5 ASMI-PROB TRAINING

To train the concentration-free version of the proposed architecture, we mostly follow Hladiš et al. (2023). However, we drop Pair imbalance weights (Eqn. (5) in (Hladiš et al., 2023)) as these weights are related to the *in vitro* exploration of protein-molecule pairs and we observe that these weights lower the performance. We also experimented with weights based on the receptor broadness (Lalis et al., 2024b) (data not shown), but their gain was not significantly different compared to standard class imbalance weights. We train ASMI-Prob for 2500 epochs with the batch size 1024 and the initial learning rate $\frac{1}{\sqrt{256}}$. The best epoch was chosen based on the validation set comprising 10% of randomly selected pairs.

### A.3.6 ASMI-REG TRAINING

We train the regression variant of the proposed architecture similarly to ASMI-DR. We train for 1500 epochs with the batch size of 1024, the initial learning rate $\frac{1}{\sqrt{256}}$, 6000 warm-up steps, and graph padding of 32 and 64 for nodes and edges, respectively. We use $l_2$ loss to train the model. For inactive pairs, we set training $EC_{50}$ values to 1M. As before, the best epoch was chosen based on the validation set comprising 10% of randomly selected pairs.

## A.4 DATA PREPROCESSING

### A.4.1 M2OR

M2OR gathers 77611 experiments, some of which correspond to the same protein-molecule pairs, and 71454 are screening data. During preprocessing, we first discard mixtures of molecules and experiments measuring the basal activity (i.e., experiments performed at molecular concentration of 0M). We also discard data reporting "fold" and "micro-ampere" units. We further remove the data with inconsistent activity decisions between experiments. For screening data, we remove experiments where, for a given protein-molecule pair, the activity decision in the higher concentration is "inactive" and in a lower concentration it is "active". We exclude galaxolide due to its unspecific response (Lalis et al., 2024b) and we also exclude molecules with molecular graphs composed of more than 32 nodes or 64 directed edges. In addition, we exclude putative pseudo genes with sequence length lower than 296 amino acids. Finally, we change the units to $log_{10}(mM)$ and if there are multiple dose-response experiments for a given pair, we take the mean of the $EC_{50}$ values. After preprocessing, the dataset consists of 1427 $EC_{50}$ values for active pairs, 4346 inactive dose-response pairs, and 60256 screening samples.

### A.4.2 DAVIS

We downloaded the curated data from PyTDC library (Huang et al., 2021). Since PyTDC does not provide activity label, and for consistency with the previous work, we obtain the labels from the preprocessed data from HyperAttentionDTI publication (Zhao et al., 2021). We exclude 6 protein sequences longer than 1736 amino acids, and we exclude molecular graphs with more than 128 nodes or 256 directed edges. Finally, we changed the units to $log_{10}(\mu M)$. After preprocessing, the data contains 6881 active and 17667 inactive pairs.

### A.4.3 BINDINGDB

We downloaded the curated data from PyTDC library (Huang et al., 2021). If there are duplicated experiments for the same pair, we take the lowest $K_d$. After analysing the data distribution, and in line with standard practeces (Huang et al., 2021), we consider pairs with $K_d \geq 10\mu M$ as inactive. We exclude 12 protein sequences longer than 2048 amino acids, and we exclude molecular graphs with more than 128 nodes or 256 directed edges. Finally, we changed the units to $log_{10}(\mu M)$. After preprocessing, the data contains 19233 active and 23001 inactive pairs.

### A.4.4 KIBA

We downloaded the preprocessed dataset from HyperAttentionDTI publication (Zhao et al., 2021). We exclude 2 protein sequences longer than 1408 amino acids, and we exclude molecular graphs

with more than 128 nodes or 256 directed edges. After preprocessing, the data contains 22154 active and 94195 inactive pairs.

## A.5 EVALUATION DETAILS

### A.5.1 *In vitro* $EC_{50}$ ERROR

Measuring $EC_{50}$ *in vitro* is prone to errors arising from several different sources (Malo et al., 2006; Brideau et al., 2003). In this work, this error is estimated by root mean squared logarithmic error (RMSLE) based on the protein-molecule pairs for which more than one dose-response experiment has been conducted. Formally, the error is given by the differences between individual $EC_{50}$ values and the mean $EC_{50}$ corresponding to the same protein-molecule pair:

$$\text{Experimental RMSLE} = \sqrt{\frac{1}{L} \sum_{k=1}^{K} \sum_{i=1}^{I_k} (x_{i,k} - \mu_k)^2} \tag{6}$$

where $K$ is the number of protein-molecule pairs with multiple $EC_{50}$ experiments, $I_k$ is the number of experiments per pair $k$, $x_{i,k}$ is the individual $EC_{50}$ value $i$ for pair $k$ in logarithmic scale, $\mu_k = \sum_{i=1}^{I_k} x_{i,k}$ is the mean of the experiments for pair $k$ and $L = \sum_{k=1}^{K} I_k$ is the total number of experiments for protein-molecule pairs with multiple $EC_{50}$ values in the dataset.

### A.5.2 DOCKING AND BOLTZ-2 DETAILS

**Docking protocol.** Gypsum-DL (v1.2.0) was used to generate 3D molecular structures from SMILES to SDF, accounting for ionisation, tautomeric, chiral, cis/trans, and ring-conformational states at pH 7.0 ± 0.5. Structures were converted from SDF to MOL2 with Open Babel 3.1.0 and then to PDBQT using MGLTools (v1.5.7). Docking was performed with SMINA (Oct 15, 2019, based on AutoDock Vina 1.1.2) (Masters et al., 2020) using the Vinardo scoring function (Quiroga & Villarreal, 2016) and an exhaustiveness of 8. Olfactory receptor models were obtained from the AlphaFold DB (Varadi et al., 2023; 2021). Polar hydrogens were added with PDB2PQR (v3.6.1), with protonation states assigned by PROPKA (v1.0) at pH 7.0, followed by minimisation with the AMBER99 force field. Protein structures were converted from PQR to PDBQT using MGLTools, and all receptors were superimposed with PyMOL (v2.5.4).

**Boltz-2 inference.** Olfactory receptor sequences and molecular SMILES were obtained from the M2OR database. We performed the experiments with Boltz-2 default parameters except: the number of diffusion samples was set to 25 for both the structure and affinity modules, and the number of recycling steps was set to 10.

## A.6 EVALUATION ON DRUG-TARGET INTERACTION DATASETS

To validate our architectural choices in Section 5, we evaluate a concentration-free version of the model, ASMI-Prob, on two standard drug-target interaction (DTI) benchmarks: KIBA (Tang et al., 2014) and DAVIS (Davis et al., 2011). We compare our architecture against state-of-the-art methods, including MolTrans (Huang et al., 2020), HyperAttentionDTI (Zhao et al., 2021).

As shown in Table A2, our architecture outperforms MolTrans in AveP on both benchmarks. It lags behind HyperAttentionDTI by $2.1\%$ and $1.9\%$ on KIBA and DAVIS, respectively, with a superior precision, whereas HyperAttentionDTI exhibit a higher recall. However, our proposed architecture excels in a challenging M2OR dataset, outperforming the DTI baselines by a margin. Notably, MCC of ASMI-DR trained by Algorithm 1 also surpasses all the DTI baselines on DAVIS dataset. This result demonstrates that our architecture is not only effective for its primary OR-molecule activation task but also robust enough to compete with the state-of-the-art on standard DTI benchmarks.

## A.7 SAMPLING WITH $EC_{50}$ LOWER BOUND AND SCREENING DATA

**Screening.** Before running demanding dose-response experiments, a screening is performed on a large number of protein-molecule pairs to assess the plausibility of the tested compounds to be ligands (Saito et al., 2009; Geithe et al., 2015; Yasi et al., 2019). The candidate molecules are first

Table A2: Performance on the KIBA and DAVIS drug-target interaction benchmarks and on M2OR. For M2OR, the results for MAARDTI are taken from Zhan et al. (2025) and the results for MolTrans, HyperAttentionDTI, and GNN-CLS are taken from Hladiš et al. (2023). Standard deviation is given in parentheses.

| | Model | AveP | Precision | Recall | MCC |
|---|---|---|---|---|---|
| DAVIS | MolTrans | 0.798 (0.017) | 0.593 (0.041) | 0.857 (0.023) | 0.584 (0.022) |
| | HyperAttentionDTI | **0.844** (0.008) | 0.767 (0.014) | 0.774 (0.016) | 0.680 (0.006) |
| | GNN-CLS | 0.744 (0.015) | 0.699 (0.013) | 0.666 (0.016) | 0.564 (0.011) |
| | ASMI-Prob | 0.828 (0.028) | 0.790 (0.012) | 0.680 (0.028) | 0.639 (0.023) |
| | ASMI-Reg | 0.817 (0.026) | 0.646 (0.063) | 0.818 (0.030) | 0.601 (0.048) |
| | ASMI-DR | 0.808 (0.015) | 0.772 (0.016) | 0.775 (0.017) | **0.685** (0.005) |
| KIBA | MolTrans | 0.767 (0.026) | 0.501 (0.032) | 0.875 (0.009) | 0.583 (0.030) |
| | HyperAttentionDTI | **0.819** (0.004) | 0.718 (0.008) | 0.770 (0.005) | **0.680** (0.005) |
| | GNN-CLS | 0.704 (0.010) | 0.610 (0.014) | 0.683 (0.005) | 0.555 (0.012) |
| | ASMI-Prob | 0.802 (0.012) | 0.738 (0.024) | 0.655 (0.059) | 0.628 (0.034) |
| M2OR | MolTrans | 0.638 (0.066) | 0.402 (0.053) | 0.822 (0.027) | 0.476 (0.042) |
| | MAARDTI | 0.700 | 0.700 | 0.595 | 0.555 |
| | HyperAttentionDTI | 0.737 (0.015) | 0.609 (0.028) | 0.773 (0.020) | 0.584 (0.022) |
| | GNN-CLS | 0.780 (0.012) | 0.689 (0.016) | 0.698 (0.042) | 0.605 (0.017) |
| | ASMI-Prob | **0.801** (0.044) | 0.716 (0.065) | 0.728 (0.037) | 0.625 (0.036) |
| | ASMI-Reg | 0.698 (0.010) | 0.692 (0.047) | 0.720 (0.035) | 0.621 (0.048) |
| | ASMI-DR | 0.754 (0.018) | 0.773 (0.028) | 0.722 (0.051) | **0.671** (0.016) |

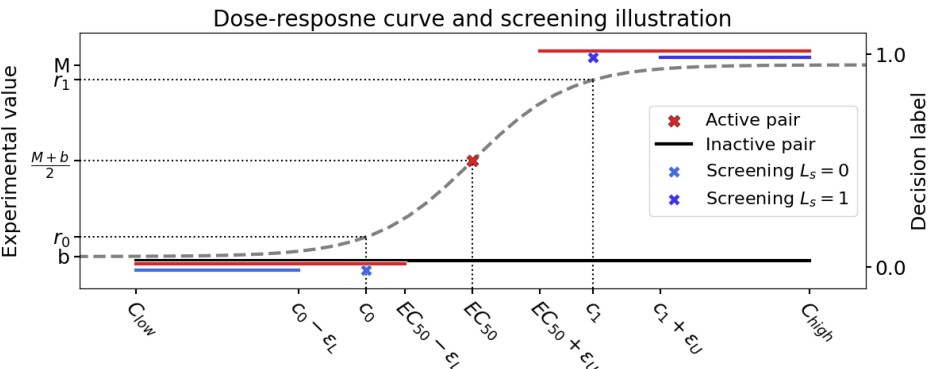

Figure A4: Example of a dose-response curve for an active (grey/red) and inactive pair (black). The crosses represent the data available in the dataset, the coloured lines are sampling regions, and the grey line is an unknown actual active curve. The blue lines correspond to screening data, the red lines are sampling regions for active $EC_{50}$ data, and the black line is the sampling region for inactive $EC_{50}$ data. $r_0$, $r_1$, $b$, $M$, and $\frac{M+b}{2}$ are experimentally measured responses which are not available in the data.

tested at a single concentration in a primary screening, followed by a secondary screening with 2 to 4 different concentrations. Screening can lead to a significant label noise (Lalis et al., 2024b), but can cover a large number of protein-molecule pairs, and considering its lower price, it constitutes the majority of the available data.

In Section 4, we consider sampling training data $\mathcal{B} = \{(s^i, m^i, c^{i,j}), L^{i,j}\}_{i,j}$ using dose-response experiments only. However, our framework allows for a straightforward extension of the same sampling procedure to the abundant screening data. Since several normalisation procedures are followed in the screening data treatment, we assume that a screening experiment results in a binary decision about whether an interaction between a protein and a molecule at a given concentration

Table A3: Sampling regions for all available data types and all possible training labels. $\epsilon_L$, $\epsilon_U$ are lower and upper margins, respectively. The case $EC_{50} > c_t$ stands for $EC_{50}$ experiments where activity was observed, but the $EC_{50}$ is out of range of the tested concentrations, and the authors reported the lower bound $c_t$ for the $EC_{50}$ value.

| Data type | Decision | $L^{i,j} = 0$ | $L^{i,j} \sim Unif(0,1)$ | $L^{i,j} = 1$ |
|---|---|---|---|---|
| $EC_{50}$ | active | $[C_{low}, EC_{50} - \epsilon_L]$ | $(EC_{50} - \epsilon_L, EC_{50} + \epsilon_U)$ | $[EC_{50} + \epsilon_U, C_{high}]$ |
| $EC_{50}$ | inactive | $[C_{low}, C_{high}]$ | $\emptyset$ | $\emptyset$ |
| $EC_{50} > c_t$ | active | $[C_{low}, c_t - \epsilon_L]$ | $(c_t - \epsilon_L, C_{high}]$ | $\emptyset$ |
| Screening | active | $\emptyset$ | $[C_{low}, c_t + \epsilon_U)$ | $[c_t + \epsilon_U, C_{high}]$ |
| Screening | inactive | $[C_{low}, c_s - \epsilon_L]$ | $(c_s - \epsilon_L, C_{high}]$ | $\emptyset$ |

elucidates the response in the cell. For secondary screening, decisions at multiple concentrations are available, and we treat them as multiple primary screenings.

Consider that a concentration $c_s$ has been tested with the corresponding response decision $L_s \in \{0, 1\}$. If the experimental response decision is "inactive" ($L_s = 0$), the monotonicity of the dose-response curve implies that for all lower concentrations $c \leq c_s$ the response decision would also be inactive (Figure A4). Analogously, if the response decision is $L_s = 1$, all higher concentrations $c \geq c_s$ would lead to the "active" decision. Therefore, "one-sided" training samples and their corresponding response decisions can be obtained for each protein-molecule pair $i$, by uniformly sampling a concentration $c^{i,j} \in [C_{low}, C_{high}]$ and then sampling a decision $L^{i,j}$ according to the experimental decision. If $L_s = 0$, then for $c^{i,j} \leq c_s$ we set $L^{i,j} = 0$ and for $c^{i,j} > c_s$ we uniformly sample a soft label $L^{i,j} \sim Unif(0,1)$. Similarly, if $L_s = 1$, then for $c^{i,j} \geq c_s$ we set $L^{i,j} = 1$ and we sample a soft label $L^{i,j} \sim Unif(0,1)$ for $c^{i,j} < c_s$. Screening can lead to label noise, and the true activity might be different from the decision available in the data. To further control the sampling, we take into account the uncertainty about the label by margins $\epsilon_L$ and $\epsilon_U$. A summary of all possible sampling regions is given in Table A3.

**$EC_{50}$ lower bound.** It is possible that an increase in the response has been observed in a dose-response assay, but the $EC_{50}$ is outside the range of the tested concentrations. Thus, the curve cannot be fitted and only a lower bound $c_t$ of the $EC_{50}$ is available. In such cases, negative samples can still be drawn from the dose-response curve, and the sampling is analogous to the negative screening case.

### A.7.1 PERFORMANCE WITH SCREENING DATA

We report the performance of incorporating the screening data in the activity decision and $EC_{50}$ estimation tasks in Table A4. For out-of-distribution (OOD) evaluation, we contrast a standard generalisation task on unseen proteins and molecules without any prior information (*all w/o screening*) against a scenario where the screening data of the test set pairs are available to the model during training (*all w/ screening*). Furthermore, we also report the performance of a model trained only on the screening data in an i.i.d. case (*screening only*).

To reflect the difference in label uncertainty of the screening and dose-response experiments, we oversample the dose-response data. For evaluation in Table A4, we sample 15 concentrations from each dose-response experiment in each epoch. Note that the test sets in Table A4 are identical to those in Section 7 and contain only dose-response assay data, excluding screening.

Although the screening data constitutes $91\%$ of the preprocessed M2OR, its integration in Algorithm 1 lowers the activity decision performance. In the OOD case, the $EC_{50}$ estimation error decreases slightly compared to the training without screening, but these results are evaluated on less pairs due to the lower MCC. Using only the screening data substantially lags behind the training with dose-response experiments. However, it still achieves an $EC_{50}$ estimation error of 0.95 log units on the correctly predicted active pairs, outperforming Boltz-2 in both MCC and RMSLE.

Overall, while screening can cover a substantial number of protein-molecule pairs, it provides less information compared to dose-response experiments. Screening data only allows for "one-sided" sampling in Algorithm 1, which leads to a severe label imbalance, especially for concentrations close

Table A4: Comparison of the performance when trained with the dose-response data only (*ASMI-DR*), all data including screening (*all*), and the screening data only (*screening only*). In the out-of-distribution evaluation, we either report a case when the screening data of the test set pairs are available to the model during training (*all w/ screening*) or a standard case when no information about the test set pairs is available (*all w/o screening*). Standard deviation is given in parentheses.

| Datacase | | Name | MCC ↑ | Precision ↑ | RMSLE ↓ | Spearman's $\rho$ ↑ |
|---|---|---|---|---|---|---|
| Primary sc. | | | 0.238 | 0.563 | | |
| Secondary sc. | | | 0.476 | 0.704 | | |
| $EC_{50}$ error | | | | | 0.334 | |
| i.i.d. | | Mean model | | | 0.899 (0.025) | |
| | | Boltz-2 | 0.108 (0.033) | 0.541 (0.117) | $1.110^{a}$ (0.037) | 0.148 (0.052) |
| | | ASMI-Reg | 0.621 (0.048) | 0.692 (0.047) | 1.213 (0.135) | 0.399 (0.122) |
| | | ASMI-DR | **0.671** (0.016) | 0.773 (0.028) | **0.725** (0.070) | **0.648** (0.065) |
| | | all | 0.652 (0.028) | | 0.772 (0.120) | 0.553 (0.148) |
| | | screening only | 0.134 (0.103) | | 0.950 (0.187) | 0.077 (0.138) |
| Sequence | Single | ASMI-Reg | 0.398 (0.112) | 0.506 (0.142) | 1.543 (0.512) | 0.150 (0.281) |
| | | ASMI-DR | **0.481** (0.031) | 0.642 (0.038) | 0.761 (0.150) | 0.470 (0.119) |
| | | all w/o screening | 0.358 (0.088) | | 0.755 (0.085) | **0.531** (0.142) |
| | | all w/ screening | 0.284 (0.090) | | **0.749** (0.042) | 0.469 (0.068) |
| | Cluster | ASMI-Reg | **0.238** (0.123) | 0.362 (0.113) | 1.889 (0.481) | -0.145 (0.158) |
| | | ASMI-DR | 0.218 (0.043) | 0.461 (0.162) | 1.170 (0.269) | 0.040 (0.150) |
| | | all w/o screening | 0.012 (0.060) | | 1.014 (0.312) | 0.127 (0.159) |
| | | all w/ screening | 0.083 (0.098) | | **0.922** (0.152) | **0.277** (0.148) |
| Molecule | Single | ASMI-Reg | 0.531 (0.054) | 0.572 (0.086) | 1.729 (0.347) | 0.286 (0.152) |
| | | ASMI-DR | **0.593** (0.074) | 0.663 (0.098) | 0.920 (0.096) | **0.474** (0.116) |
| | | all w/o screening | 0.469 (0.085) | | 0.916 (0.070) | 0.406 (0.035) |
| | | all w/ screening | 0.547 (0.060) | | **0.895** (0.152) | 0.428 (0.137) |
| | Cluster | ASMI-Reg | 0.395 (0.082) | 0.548 (0.151) | 1.561 (0.371) | 0.154 (0.107) |
| | | ASMI-DR | **0.398** (0.077) | 0.572 (0.115) | 0.818 (0.154) | **0.298** (0.116) |
| | | all w/o screening | 0.309 (0.091) | | 0.798 (0.112) | 0.154 (0.135) |
| | | all w/ screening | 0.227 (0.065) | | **0.748** (0.044) | 0.274 (0.114) |

[a]Due to the low MCC, the evaluation is also done on incorrectly classified pairs.

to the boundaries of the sampling region. Indeed, if screening is performed at a high concentration $c_s \approx C_{high}$, then without access to the inactive dose-response data, which provides inactive samples $\{(s^i, m^i, c^{i,j}), L^{i,j} = 0\}$ for all $c^{i,j} \in [C_{low}, C_{high}]$, the model only has access to active examples at $c_s \approx C_{high}$. In future work, we aim to address this limitation by adjusting the sampling strategy for the screening data.

## A.8  $EC_{50}$ ORDER EVALUATION

According to the primacy coding theory in olfaction (Wilson et al., 2017; Zwicker, 2019), the order of activation of olfactory receptors plays a pivotal role in the odour perception of a molecule. To assess the ability of the model to assign the rank, we evaluate the mean Spearman's rank correlation for individual molecules and proteins in Table A5. While the rank correlation per protein reaches a level comparable to the correlation among all the pairs (*ASMI-DR* in Table A5), the correlation per molecule is $40\%$ lower in the i.i.d. case. Notably, the model achieves a higher $\rho$ per molecule for novel molecules, but it fails to predict the order of activation per molecule when considering a new protein sequence.

Table A5: $EC_{50}$ order evaluation. Spearman's $\rho$ is the rank correlation between the $EC_{50}$ estimated from the ASMI-DR model predictions and the experimentally measured values. Rows *per molecule* and *per protein* correspond to the mean of correlation per molecule and per protein, respectively. The standard deviation of 5 runs is given in parentheses.

| Datacase | | Name | MCC ↑ | Precision ↑ | RMSLE ↓ | Spearman's $\rho$ ↑ |
|---|---|---|---|---|---|---|
| i.i.d. | | ASMI-Reg | 0.621 (0.048) | 0.692 (0.047) | 1.213 (0.135) | 0.399 (0.122) |
| | | ASMI-DR per molecule per sequence | **0.671** (0.016) | **0.773** (0.028) | **0.725** (0.070) | **0.648** (0.065) 0.319 (0.225) 0.528 (0.052) |
| Sequence | Single | ASMI-Reg | 0.398 (0.112) | 0.506 (0.142) | 1.543 (0.512) | 0.150 (0.281) |
| | | ASMI-DR per molecule per sequence | **0.481** (0.031) | **0.642** (0.038) | **0.761** (0.150) | **0.470** (0.119) -0.134 (0.063) 0.464 (0.142) |
| | Cluster | ASMI-Reg | **0.238** (0.123) | 0.362 (0.113) | 1.889 (0.481) | -0.145 (0.158) |
| | | ASMI-DR per molecule per sequence | 0.218 (0.043) | **0.461** (0.162) | **1.170** (0.269) | **0.040** (0.150) 0.167 (0.357) 0.151 (0.243) |
| Molecule | Single | ASMI-Reg | 0.531 (0.054) | 0.572 (0.086) | 1.729 (0.347) | 0.286 (0.152) |
| | | ASMI-DR per molecule per sequence | **0.593** (0.074) | **0.663** (0.098) | **0.920** (0.096) | **0.474** (0.116) 0.244 (0.144) 0.290 (0.134) |
| | Cluster | ASMI-Reg | 0.395 (0.082) | 0.548 (0.151) | 1.561 (0.371) | 0.154 (0.107) |
| | | ASMI-DR per molecule per sequence | **0.398** (0.077) | **0.572** (0.115) | **0.818** (0.154) | **0.298** (0.116) 0.246 (0.244) 0.150 (0.170) |

