# OpenReview forum: "From Regression to Dose–Response: A Framework to predict Activity and $EC50$ for GPCRs"
_ICLR.cc/2026/Conference — Submitted to ICLR 2026_

### Official Review · Reviewer_pBeT · 2025-10-20

**Soundness:** 3
**Presentation:** 2
**Contribution:** 3
**Rating:** 4
**Confidence:** 2

**Summary:**

This paper introduces a novel framework for predicting EC50 of G-protein coupled receptors (GPCRs).  Instead of treating EC50 prediction as a direct regression problem, the authors reformulate it as a series of binary classification tasks that mimic in vitro dose-response experiments. The core idea is to train a model that predicts the activation probability of a protein-molecule pair at a given concentration.

**Strengths:**

1. The idea that reformulating the EC50 prediction to a series of binary classification tasks is interesting.

2. Experimental results show that the proposed framework outperforms conventional direct regression approaches for EC50 prediction.

**Weaknesses:**

1. The validation is primarily conducted on the M2OR dataset. Additional protein targets or datasets should be included to assess the generalizability of the framework, such as BindingDB.

2. The study should include benchmarks against more advanced drug–target interaction prediction methods such as DTIAM[1] and GraphBAN[2].

3. The inference process is inherently more computationally expensive than direct regression. To predict the EC50 for a single protein-molecule pair, the model requires multiple queries across different concentration data points.

4. Lack of deeper insights into why the proposed framework yields better results than direct EC50 prediction. More experiments or case analyses are needed to uncover the underlying mechanisms.

[1]. Lu, Z., Song, G., Zhu, H. et al. DTIAM: a unified framework for predicting drug-target interactions, binding affinities and drug mechanisms. Nat Commun 16, 2548 (2025). https://doi.org/10.1038/s41467-025-57828-0

[2]. Hadipour H, Li Y Y, Sun Y, et al. GraphBAN: An inductive graph-based approach for enhanced prediction of compound-protein interactions[J]. Nature Communications, 2025, 16(1): 2541.

**Questions:**

1. Is the proposed EC50 prediction framework applicable to any DTI prediction model? It would be interesting to see if the benchmarked models also benefit from this.

2.  Can the framework be directly applied to IC50 prediction tasks? If validated, this would demonstrate the broader utility and robustness of the approach across different bioactivity endpoints.

3. The choice of threshold $\varepsilon_L$ and $\varepsilon_U$ is not sufficiently justified. It would be important to analyze how varying $\varepsilon_L$ and $\varepsilon_U$ affects performance, whether the optimal value differs across datasets, and if adaptive threshold selection could improve generalizability.

---

> ### Author Response · Authors · 2025-11-26
> **Part 1**
>
> We are grateful to the reviewer for the detailed evaluation and for highlighting the novelty of our proposed framework. We believe that the performance gains in both EC50 regression and activity classification tasks highlight the advantages and broad applicability of our framework and positions this work as a valuable addition to ICLR. Please find below comment addressing the weaknesses (W) and questions (Q) raise by the reviewer.
>
> ## W1
> We thank the reviewer for raising this point, and we are currently running additional evaluations for affinity prediction on standard drug-target interaction benchmarks, namely DAVIS and BindingDB, bringing the total number of datasets used in the manuscript to 4. This number also includes KIBA dataset, which can not be directly used as a regression benchmark in our approach, since KIBA score merges several types of affinity measurements and can not be seen as an inflection point of a dose-response curve. Nevertheless, we use KIBA as an evaluation dataset for the classification version of our architecture (ASMI-Prob). We will update the manuscript and report the performance in a dedicated comment as soon as possible.
>
> ## W2
> We thank the reviewer for his effort to enhance the quality of our manuscript. We are currently running additional evaluations for drug-target affinity prediction (DTA) models and for additional DTA benchmarks (DAVIS and BindingDB), and we will post the results in a dedicated comment as soon as they are ready.
>
> ## W3
> We thank the reviewer for pointing this out, and we agree that the model is inherently more computational expensive. We include a short paragraph in the limitations part in the conclusion about the computational demands. Nonetheless, the computational costs are still lower compared to structure-based approaches such as Boltz-2, and we believe that the performance gains of our approach surpass the additional computational demands.
>
> ## W4
> We thank the reviewer for raising a point on insights into the mechanism of our framework. In the response to reviewer 4Ycs (W1, W2, Q3), we outline several advantages of our approach which shed light on our understanding of the performance gains compared to separate regression and classification tasks. Briefly, as reviewer 4Ycs also pointed out, we expect that modelling concentration explicitly as an input has a regularizing effect on the network. This is in line with a classical observation in deep learning that data augmentation and perturbations in model training are beneficial for the performance.
>
> Secondly, our framework is closely related to the underlying data generation process in the wet lab experiments. A standard procedure in the wet lab is to test the response of a protein-molecule pair at several concentrations, and then fit a logistic model to the measurements spanning several orders of magnitude in concentration. Because our framework follows the same procedure, it can avoid pitfalls of oversimplification that is necessary to train regression or classification models. The golden standard in training a regression model is to set a high-enough constant as the affinity target for inactive pairs (e.g. in DAVIS dataset) and to treat censored data by simply removing ">" and using the censoring thresholds as targets (e.g. this is done in widely used PyTDC library for BindingDB). In contrast, our approach seamlessly enables to model these cases by exploiting only the information in the tested range of concentrations, without the need to extrapolate the results of experiments to assign a single value to an entire protein-molecule pairs.
>
> ## Q1
> We thank the reviewer for opening the discussion about the broad applicability of our framework, which was also pointed out by reviewer 4Ycs (W4). We specifically designed the experiments in Table 1 and Table 2 to test whether using similar architectural choices in regression, classification and dose-response (ours) training leads to a different performance. As can be seen in the results, despite having as close as possible architectures, ASMI-DR outperforms its regression (ASMI-Reg) and classification (ASMI-Prob) variants by a large margin (40\% in i.i.d regression evaluation and 10\% in i.i.d classification evaluation) in all tested scenarios. Therefore, in light of these strong results, we believe that our proposed framework is independent of the proposed ASMI architecture, and we expect performance gains for the existing DTI architectures, when adjusted for the additional concentration input.

---

> ### Author Response · Authors · 2025-11-26
> **Part 2**
>
> ## Q2
> We thank the reviewer for this interesting question. Our framework is flexible, and it only assumes the general shape of the dose-response curve (i.e. generalized logistic curve) and access to the inflection point (e.g. EC50, Kd, Ki, IC50). Thus, by simply substituting probability of activation $P(active|s,m,c)$ by probability of inhibition $P(inhibition|s,m,c)$ in line 197, the framework can be used as is for IC50 prediction. To extend the experiments about the applicability of our framework, we are currently training on DTA benchmarks, DAVIS and BindingDB, and we will report the results in the dedicated comment as soon as possible.
>
> ## Q3
> As discussed in the response to reviewer mosk (Q1), during development of our framework we tested several values of $\epsilon_{L}$ and $\epsilon_{U}$. We did not run a systematic evaluation of possible values, however, we noticed that the choice of the margins primarily affects the estimate of a slope parameter $q$ and have a minor effect on the EC50 estimation performance. In particular, adjusting the margins may enable the practitioners to incorporate domain knowledge about the expected slope of the dose-response curve.
>
> We would like to add that we consider the reviewer's suggestion to introduce adaptive margins to be an interesting future direction that may lead to enhanced performance. One advantage of such an approach is that margins could be modelled as a function of the experimental setting and can adaptively change based on the noise from the wet lab experiments.  Together with the development of an enhanced sampling strategy for screening data outlined in the discussion with reviewer 4Ycs (W1, W2, Q3), we believe that the flexibility of our proposed framework will open new avenues in modelling the complex protein-molecule interaction assays.

---

> ### Author Response · Authors · 2025-12-03
> **Update on additional evaluation**
>
> ## Update on W1 and W2
> We would like to once again thank the reviewer for his effort to enhance the quality and scope of our work and we would like to update the reviewer on the additional experiments on two drug-target affinity datasets, DAVIS and BindingDB, and a comparison to three state-of-the-art drug-target affinity prediction models.
>
> As can be seen in the dedicated comment, and in Table 2 and Table 3 in the revised manuscript, ASMI-DR outperform all previous approaches on M2OR by a large margin of more than $40\%$ (0.488 log-units). On DAVIS and BindingDB, ASMI-DR's Kd estimation performance is statistically indistinguishable from the best-performing regression baseline ProSmith  (paired t-test p-values: 0.106 and 0.315, respectively), while consistently outperforming all competing methods on the activity decision metrics (MCC and precision).
>
> We believe that the results of these additional experiments serve as a strong support in favour of our proposed framework, and that the reviewer would consider increasing his score and the level of confidence.

---

### Official Review · Reviewer_mosk · 2025-10-27

**Soundness:** 4
**Presentation:** 3
**Contribution:** 3
**Rating:** 6
**Confidence:** 3

**Summary:**

This paper introduces a novel framework that reframes EC50 prediction from a direct regression task to a series of concentration-dependent classification tasks, mimicking in vitro dose-response assays. The proposed model, ASMI-DR, learns to predict activation probability at a given concentration. By querying the model across a range of concentrations, a curve is fitted to derive both activity and EC50, reportedly achieving state-of-the-art results on the M2OR dataset.

**Strengths:**

1. The core idea of reformulating EC50 regression as a classification task that mimics biological assays is highly innovative and provides a simple but effective framework for predicting both activity and potency.
2. The paper demonstrates strong performance with a rigorous evaluation across multiple out-of-distribution scenarios and a wide range of baselines. The paper even conducted a quantitative comparison against a real-world in vitro screening workflow, which adequately demonstrates the model's practical value.

**Weaknesses:**

1. The model imposes strict requirements on its training data, which can limit its practical applicability. The framework relies on high-quality, complete dose-response curve data, which are more expensive to acquire and are scarce in existing databases (e.g., ChEMBL, BindingDB), to generate its training labels.
2. As shown in this paper, the authors admit that the framework fails when attempting to incorporate abundant single-point screening data.

**Questions:**

1. The sampling margins and soft label distribution in the training algorithm seem empirical. Could you provide a justification for these choices or a sensitivity analysis?
2. Why does the framework fail when handling abundant screening data? Have you considered alternative sampling strategies or loss functions to address this limitation?

---

> ### Author Response · Authors · 2025-11-26
>
> We thank the reviewer for the thorough evaluation of the manuscript and for recognizing the novelty of the proposed framework. We believe that the flexibility of our approach and its strong performance in both activity classification and EC50 prediction demonstrated in-distribution and out-of-distribution will be an appreciated contribution to ICLR. Please find below responses to the raised weaknesses (W) and questions (Q).
>
> ## W1, W2
> We thank the reviewer for bringing up the crucial point of data requirements, which posses a major constraint in protein-molecule interaction prediction. A major advantage of our framework is that we only assume access to the inflection point of the dose-response curve (i.e. EC50) and we consider that the rest of the curve is unknown. Indeed, while full dose-response curve data are unavailable, the data needed to train ASMI-DR is available in data sources such as BindingDB and ChEMBL and we are running additional experiments on DAVIS and BindingDB datasets at the moment.
>
> In addition, we designed our approach to be as general as possible when it comes to data sources, and we discuss in Section A.7 the possibility to train the model on abundant screening data. We consider this to be a valuable feature of our approach, as it enables a novel view on modelling data quality in protein-molecule interaction prediction. As discussed in the response to reviewer fFHL, our approach mitigates one source of the label noise coming from a particular choice of the screening concentration, enabling a unified modelling of preliminary screening assays and confirmatory dose-response experiments. While the results with the screening data in Section A.7 did not show better performance than using only EC50 values, we believe that adjusting the screening activity decision sampling strategy in our framework is a promising future direction which may enable a principled way to extract information from the abundant screening experiments.
>
> ## Q1
> We thank the reviewer for raising the point about the hyperparameter choice for activity label sampling.
>
> The choice of soft labels was based on the assumption that we have no information about the activity at concentrations around the EC50 values or when censored data are given (Section A.7, EC50 lower bound). Therefore, we chose to use an uninformative uniform distribution for soft labels. Alternatively, one could also use a fixed value of $0.5$ instead, but we decided to introduce random label perturbation to the training, as the use of data augmentation is generally considered to be beneficial for training deep learning models.
>
> During model development, we tested several values of $\epsilon$ margins, as well as having a different set of margins for the screening data in Section A.7. However, we did not run a systematic evaluation of possible values of the margins. We noticed, in these incomplete set of experiments, that the choice of $\epsilon$ margins does not have a strong influence on the EC50 prediction accuracy. We expect that the choice of $\epsilon_{L}$ and $\epsilon_{U}$ is particularly important if one wants to have a good fit of the slope parameter $q$ of a normalized dose-response curve.
>
> In general, we believe that the choice of the margins and the label distribution is a promising future extension of our framework. In particular, as we discussed in the answer to reviewer 4Ycs, adjusting the label distribution for screening data is a promising direction to model the inherent screening label noise, and it may open a new avenue for an efficient integration  of abundant screening data in the training of EC50 (or affinity) prediction models.
>
> ## Q2
> We thank the reviewer for the question, and we consider the possibility to incorporate screening data as an interesting implication of our framework. As discussed in the response to reviewer 4Ycs, we believe that the main reason for the deteriorated performance when screening data are incorporated in the training is the label noise of active protein-molecule pairs. While our approach mitigates the label noise coming from missed active pairs when a too low screening concentration is tested, it can not mitigate the noise from the plate-well effects or from unspecific response due to high concentration of the candidate molecule. Therefore, the label noise in the active screening pairs (which can lead to 60\% incorrect labels for active primary screening according to Hladiš et al.) can bias the predictions at high concentrations, leading to a high number of false negative pairs and a low MCC. We believe that an alternative sampling strategy that would treat active and inactive screening experiments differently can be developed and would bring additional performance gains to our framework. However, considering the short time window for rebuttals and several experiments that we run to evaluate our approach on additional datasets (DAVIS and BindingDB) we leave this promising direction for a future work.

---

> > ### Comment · Reviewer_mosk · 2025-11-27
> >
> > Thank you for your response. My concerns are all properly solved, and I would like to keep my positive score.

---

### Official Review · Reviewer_4Ycs · 2025-10-31

**Soundness:** 1
**Presentation:** 3
**Contribution:** 2
**Rating:** 2
**Confidence:** 3

**Summary:**

This manuscript proposes a framework for predicting EC50 values and activity
of olfactory receptors from molecular graphs and amino acid sequences.
Instead of directly predicting EC50 and activity,
the authors propose to artificially create binary classification tasks for different concentration levels,
mimicking the underlying experiments that are used to obtain these measurements, more closely.

**Strengths:**

- The idea to use concentration data for activity prediction is interesting.
 - Overall, the paper has a good reading flow and is understandable.
 - Improvements for the EC50 prediction look impressive.

**Weaknesses:**

- It is not clear how useful it is to add concentration inputs that are not available in practice.
   On line 143, the authors state that they need to produce surrogate values.
   However, since this surrogate data is obtained from the EC50 values,
   this can not add additional information and typically tends to increase noise in the data.
   It might be that this noise has a regularising effect, but it is hard to verify such hypotheses.
   If the data is obtained by performing this kind of experiments,
   I would expect that it is possible to construct a dataset of activity at different concentration values.
   This should allow to test the quality of the surrogate values and/or my regularisation hypothesis.
   Furthermore, a dataset including concentration values should make the proposed model much more powerful
   since it could directly incorporate this additional information.
 - I do not understand why the standard paradigm needs to be challenged (as stated on line 80).
   There has been evidence that multi-task learning can benefit tasks in drug discovery (e.g. Mayr et al., 2014),
   but I would have expected EC50 values to follow a Gaussian-like distribution,
   in which case regression is a sensible thing to do.
 - From my experiences, graph neural networks bring little to no benefits when working with small molecules.
   Typically, pre-computed fingerprints work just as well or better for most machine learning tasks.
   It seems like the GNN embedding network could be easily replaced by fingerprints.
 - In general, it is not clear how much of the performance comes from the novel paradigm
   and how much performance is due to the network architecture.
   An ablation study, where the SVM, RF or BiLSTM are used in the new paradigm
   might be helpful to provide some insights here.
 - There is no information on how the hyper-parameters were chosen and what alternative architectures were considered.
 - The results in Table&nbsp;1 indicate that this model is not competitive with an SVM from 2020 for activity prediction,
   even though this SVM model is likely much simpler than the proposed multi-block architecture with GNNs and cross-attention.
   This seems to indicate that this model is not going to be useful for practial activity prediction tasks.
   As a result, the claim in the abstract that the model improves state-of-the-art with 10% on activity prediction is completely unwaranted.
 - The baselines for the EC50 prediction seem to be models mainly optimised for activity prediction.
   Therefore, it is unclear how competitive these baselines are.
   Furthermore, it seems reasonable to expect that a multi-task setup enables reasonable EC50 prediction,
   however, no "plain" multi-task models have been included in the comparison.
   Simply adding a second "head" to predict the EC50 values should give a good idea.
   This would allow the SVM, RF and BiLSTM model to be included as baselines in Table&nbsp;2.
 - It is not entirely clear why the model has only been tested on olfactory receptor data.
   The method seems to be general enough to be applicable to other, more common QSAR datasets (e.g. toxicity, ...).

### minor issues
 - The main text contains plain references to figures in the appendix.
   It would be nice to make clear that these figures are not in the main part.
 - It is unclear what the small values in parentheses represent in Tables&nbsp;1 and&nbsp;2.
   Are these standard deviations, confidence intervals, IQR or something else?
 - The bold numbers in table&nbsp;1 do not make much sense.
   How is SVM not the best model in terms of Recall?

**Questions:**

1. Do you have any results applying this model to real experiment data?
 2. How well does the surrogate data model the real data?
 3. Are there any arguments to motivate the need to challenge the standard approach?
 4. How do other architectures perform when used in the new paradigm?
 5. What architectures were considered and how were the hyper-parameters chosen?
 6. What is the motivation to use graph neural networks?
 7. How well does the model perform when using molecular fingerprints instead of the GNN embeddings?
 8. What does it mean to improve SOTA with 10% if you can not match the performance of an SVM from 2020?
 9. What are the relevant metrics in Table&nbsp;1 and why?
 10. How competitive are the baselines for the EC50 prediction?
 11. How well does a simple multi-task setup work for combined activity and EC50 prediction?
 12. How does this model on more standard QSAR (e.g. toxicity) prediction tasks?
 13. What are the values in parentheses in Tables&nbsp;1 and&nbsp;2?

---

> ### Author Response · Authors · 2025-11-26
> **Part 1**
>
> We thank the reviewer for investing effort to thoroughly evaluate our work, and we appreciate that the reviewer deem our proposed framework interesting. Encouraged by the strong performance of ASMI-DR in both regression and classification tasks, we believe that our work can be a valuable contribution to the ICLR community. In the next comments, we address the weaknesses (W) and questions (Q) raised by the reviewer, and we hope that our arguments will support the acceptance of our manuscript.
>
> ## W1, W2, Q3
> We thank the reviewer for raising a crucial point on the reasoning behind the sampling strategy. This is strongly linked to the reviewer's question of "why the standard paradigm needs to be challenged" so we will combine the answers to these two points.
>
> As we discuss with reviewer fFHL who has directed us to the related work "Predicting Dose-Response Curves with Deep Neural Networks." by Alonso-Campaña et al. [1] it has been shown that it is beneficial to model raw response data instead of modelling the parameters of a dose-response curve. In our work, we generalize this observation, and we show that even if one does not have access to the raw dose-response measurements, treating the problem as activity prediction at a given concentration -- as done in the wet lab experiments -- leads to a more accurate estimate of both the activity and the inflection point of a dose-response curve (Table 1 and Table 2). We believe that one of the reasons is the regularizing effect that the reviewer mentioned and the second is that our framework follows the wet lab data generating process, and it can better model potential biases coming from this process.
>
> Firstly, we fully agree with the reviewer that "a dataset including concentration values should make the proposed model much more powerful, since it could directly incorporate this additional information". We believe that ASMI-DR would perform well in a setting with the raw dose-response data, however, such dataset is rarely available in practice, and we are not aware of any curated dataset for dose-response curves in protein-molecule interaction. Thus, we assume to only have access to the inflection point of the curves (i.e., EC50 values). In addition, as noted in the response to reviewer fFHL, different readouts and experimental settings used in the functional assays of M2OR dataset make direct comparison of the raw dose-response data non-trivial, even if they would be available.
>
> (See next comment)

---

> > ### Author Response · Authors · 2025-11-26
> > **W1, W2, Q3 - continued**
> >
> > ## W1, W2, Q3 - continued
> >
> > The second point are biases of the available data, that can be better modelled by our framework. We fully agree with the reviewer that regression is a valid modelling direction to take, which is also shown by ASMI-Reg, which lags behind Boltz by only 0.1 log units). On the other hand, regression approach lacks the flexibility of our proposed dose-response sampling in several points:
> >
> > - The treatment of non-responsive molecule-protein pairs is non-trivial in regression approach and a golden standard in drug-target affinity prediction is to set a large enough constant for inactive examples (e.g. 10 uM in DAVIS dataset). While in theory this could be learned by a regression model, it adds dificulty to the regression task, as there are two modes in the affinity/EC50 distribution (active and non-active) and the regression model needs to learn them both. In contrast, our approach seamlessly works with negative samples as it only considers decision at a given concentration and there is no need to set "dummy" inactive target values.
> >
> > - Additionally, a similar argument holds for censored data (referred to as "EC50 lower bound" in Section A.7). Currently, the method of choice in regression is to simply remove ">" sign form the censored samples and treat the threshold $c_{t}$ as a target value (this is done, for example, by the widely used PyTDC library for BindingDB). Once again, our sampling procedure treats this data by only considering that the pair is inactive below the censored threshold, but not assuming any particular label for the concentrations above the threshold (we randomly sample a soft label form $(0, 1)$ for concentrations above the threshold).
> >
> > - Lastly, despite the fact, that including the screening data in the training pipeline did not increase the performance, as shown in Section A.7, we still believe that this is a valuable feature of our approach. Screening data constitute majority of the available experiments (89\% of pairs in M2OR) and in regression, screening data can not be incorporated in the model training. Even from classification point of view, our approach mitigates one source of the label noise. The majority of primary screening experiments in M2OR are performed at 100uM ($28949$ out of $53444$ molecule-receptor pairs after preprocessing) and, as pointed out by Lalis et al. [2], this implies that 48\% of active pairs could have been missed only due to the 100uM choice of the screening concentration. In our approach, we only use the information provided by the screening experiment, and if a pair is inactive in a low concentration, we avoid setting the activity decision to "inactive" for the whole pair, as done in classification. We would like to point out, that while we believe that our approach mitigates a one source of label noise, it is not treating other sources such as plate-well effects in *in vitro* experiments [3, 4]. We believe that this is the main reason for the deteriorated results in Section A.7, where the label noise in "active" screening experiments leads to inaccurate decisions at high concentrations (precision of primary and secondary screening is only 40\% and 72\%, respectively, according to Hladiš et al. [5]). We believe that tackling the label noise in the scope of our framework is a promising future direction in modelling the abundant screening experiments.

---

> ### Author Response · Authors · 2025-11-26
> **Part 2**
>
> ## W3, Q6, Q7
> We agree with the reviewer that there is no "go-to" molecular representation in protein-molecule interaction literature and models use a variety of approaches, including graph neural networks (GNNs), molecular fingerprints, or SMILES-based representation from pretrained Transformer models. In this work, we primarily followed the previous SOTA on M2OR dataset (Hladš et al. [5]), which uses a tailored GNN architecture combining message passing neural networks with multi-head attention. The second reason to use GNN is that self-attention in "Sequence update block" can be seen as a GNN applied to a fully-connected graph where amino acids in the protein sequence are nodes in the graph. Following this, we designed "Node update block" analogously and used a GNN to process molecular graphs.
>
> Considering the short time window for rebuttals and many benchmarking experiments we are currently running (additional datasets and additional baselines), we are unable to run a thorough ablation of ASMI-DR using molecular fingerprints. However, we are running experiments on several baseline models which are based on other molecular representations, such as ChemBERTa. Lastly, we would like to emphasize, that we agree with the reviewer that molecular fingerprints might be an alternative choice which can be used in a subsequent work that built upon our proposed framework.
>
> ## W4, Q4
> We thank the reviewer for the remark on the gains from architecture choice versus from the novel paradigm. In our experiments in Table 1, Table 2 and Table 6, we specifically tried to disentangle this difference by using identical architectures (except concentration input) in classification (ASMI-Prob), regression (ASMI-Reg) and the dose-response paradigm (ASMI-DR). As can be seen in the results, ASMI-DR outperforms the two other approaches in both classification and regression tasks.
>
> While we also believe that additional experiments and ablations are always beneficial for the reader, we would like to point out, that we are currently running several additional experiments on binding affinity prediction datasets and other model architectures. Additionally, other model architectures were designed with either regression or classification goal, and they do not consider concentration as an input. In our experiments we invested effort to design all three versions of the architecture (ASMI-Reg, ASMI-Prob and ASMI-DR) to be as performant as possible in their task while keeping the given architectural blueprint. Thus, naively extending the current models in the short time window for rebuttals can lead to their underperformance and provide misleading results for the reader.
>
> In ML it is always possible to deign a more performant model. However, we believe that our experiments in Table 1, Table 2 and Table 6 show that the novel paradigm is the reason for the performance gains and that future models will benefit from our proposed dose-response framework, even if a different model architecture is used.

---

> ### Author Response · Authors · 2025-11-26
> **Part 3**
>
> ## W6, Q8
> We thank the reviewer for pointing out the potential confusion in the applicability of our work, and we admit that the presentation in Table 1 might be misleading for the reader. To clarify, as discussed in Hladiš et al. [5], SVM models from Kowalewski and Ray [6] take a less-challenging "molecule-oriented" approach which aims to predict active compounds for a fixed set of olfactory receptors (ORs). The authors trained a separate SVM model for each olfactory receptor, and because these models do not take protein sequence nor structure into account, they can only be applied to ORs with enough data (the authors trained and evaluated the models only on ORs with at least 3 known active compounds, leading to the applicability domain of only 34 OR sequences). Thus, the generality of this approach is limited, particularly for olfactory receptors, for which 42\% of tested protein sequences are orphans [2], i.e. have no known active compound. We included the performance of these SVM models from the original publication [6] in Table 1 for completeness, but we agree that this might be confusing. Therefore, if the reviewer finds this appropriate, we will keep the SVMs in the related work section, and we will move the discussion about the performance of these models from Table 1 to the text. The same holds for RF from Cong et al. [7], which takes "receptor-oriented" approach and predict activated ORs for a fixed set of molecules with enough data. The authors considered 10 molecules for which they predicted active ORs in their study.
>
> We would like to note, that test sets of ASMI-DR evaluation contain $213$ and $237$ protein sequences and molecules on average, respectively, and can be applied to virtually any OR-molecule pair. Yet, despite the fact that SVM models from [6] are evaluated in a less challenging and less general task, ASMI-DR lags behind them only by 0.02 in AUROC, which largely outperforms GNN-CLS, which is the current SOTA.
>
> ## W8, Q12
> In line with other reviewers who also asked additional evaluation on standard benchmarks, we are currently running experiments on standard drug-target interaction datasets: DAVIS and BindingDB. We will update the values here in the dedicated comment and in the manuscript as soon as we have the results.
>
> ## Q9
> Similarly to other protein-molecule interaction datasets, label distribution of M2OR is imbalanced (24.7\% active ec50 pairs). Therefore, in line with the previous work of Hladiš et at. [5] we use Mathew's correlation coefficient (MCC) as the main metric. We also report other standard metrics for completeness.
>
> ## Q13
> We thank the reviewer for his remarks to enhance the quality of the manuscript for the reader. The values are standard deviations across 5 test sets, and we updated the captions of all the tables to improve the clarity.

---

> ### Author Response · Authors · 2025-11-26
> **References**
>
> [1] P. Alonso-Campaña, P. Prasse, T. Scheffer. Predicting Dose-Response Curves with Deep Neural Networks. Forty-first International Conference on Machine Learning, 2024.
>
> [2] M. Lalis, M. Hladiš, S. Abi Khalil, C. Deroo, C. Marin, M. Bensafi, N. Baldovini, L. Briand, S. Fiorucci, and J. Topin. A status report on human odorant receptors and their allocated agonists. Chemical Senses, 49:bjae037, 10 2024b. ISSN 1464-3553. doi: 10.1093/chemse/bjae037.
>
> [3] N. Malo, J.A. Hanley, S. Cerquozzi, J. Pelletier, and R. Nadon. Statistical practice in high-throughput screening data analysis. Nature Biotechnology, 24(2):167–175, 2006.
>
> [4] C. Brideau, B. Gunter, B. Pikounis, and A. Liaw. Improved statistical methods for hit selection in high-throughput screening. SLAS Discovery, 8(6):634–647, 2003.
>
> [5] M. Hladiš, M. Lalis, S. Fiorucci, and J. Topin. Matching receptor to odorant with protein language and graph neural networks. In The Eleventh International Conference on Learning Representations, 2023.
>
> [6] J. Kowalewski, A. Ray. Predicting human olfactory perception from activities of odorant receptors. iScience, 23(8):101361, 2020. ISSN 2589-0042. doi: https://doi.org/10.1016/j.isci.2020.101361.
>
> [7] X. Cong, W. Ren, J. Pacalon, R. Xu, L. Xu, X. Li, C.A. de March, H. Matsunami, H. Yu, Y. Yu, and J. Golebiowski. Large-scale G protein-coupled olfactory receptor–ligand pairing. ACS Central Science, 8(3):379–387, 2022. doi: 10.1021/acscentsci.1c01495.

---

> > ### Comment · Reviewer_4Ycs · 2025-11-26
> > **Thank you for addressing most of my concerns**
> >
> > > [in] [1] it has been shown that it is beneficial to model raw response data instead of modelling the parameters of a dose-response curve
> >
> > If I understood it correctly, this paper argues that the standard parametric model for dose-response curves is too limited.
> > However, the proposed approach in this manuscript is estimating the parameters for this standard model and therefore also limited. In that sense [1] does not contribute to the motivation in favour of this work, but rather seems to highlight inherent limitations of the proposed approach.
> >
> > > We believe that ASMI-DR would perform well in a setting with the raw dose-response data, however, such dataset is rarely available in practice
> >
> > The point is that you should experimentally verify that this actually works, to prove that this method actually does what you claim it is doing.
> > Concretely, I would expect an experiment where you know the full dose-response curve and reconstruct it (better than other methods) from the EC50 values. Following [1], there should be at least some data out there.
> > Once this has been established, I am willing to believe that this could work for data where only EC50 values are available.
> >
> > > regression approach lacks the flexibility of our proposed dose-response sampling
> >
> > In [1], regression seems to work well enough to model non-standard dose-response curves, so I would expect it to work well enough for modelling dose-response curves following the standard parametric model.
> >
> > > move the discussion about the performance of these models from Table 1 to the text
> >
> > It would definitely be better to make sure that all results in a table are at least comparable.
> > However, it might be useful to repeat the experiments from [6] and show that the proposed model at least outperforms the SVM in this simplified setting. Otherwise, it would require a very thorough discussion as to why this setting is not relevant (anymore).
> >
> > > we are currently running experiments on standard drug-target interaction datasets
> >
> > I am afraid that I am not inclined to accept a paper based on results of a set of completely new experiments that are hastily performed in the course of a rebuttal.
> >
> > Overall, I think the paper was not ready for submission and I am afraid that the rebuttal period will not provide enough time to properly rework this manuscript to a full-fledged paper. Especially, given that most results are not available and crucial related work was not taken into account. However, assuming that results on the standard benchmarks turn out well and comments from the current reviews are taken into account, it could be valuable to resubmit this work to future conferences.

---

### Official Review · Reviewer_fFHL · 2025-11-01

**Soundness:** 2
**Presentation:** 3
**Contribution:** 2
**Rating:** 4
**Confidence:** 4

**Summary:**

The paper presents a new framework for predicting GPCR activity and EC50 by mimicking in vitro dose-response assays, thereby capturing the concentration-dependent nature of receptor activation. The authors introduce ASMI-DR, a model that estimates activation probabilities across concentrations, fits a logistic curve, and derives both activity and EC50 from it. ASMI-DR achieves strong performance compared to existing baselines and demonstrates robust generalization to novel molecular scaffolds and receptor variants.

**Strengths:**

1. The core idea of modeling dose–response relationships through this framework is intuitive and well‑motivated.
2. The experiment results look promising.
3. The method is clearly written and well-delivered.

**Weaknesses:**

1. Several baselines are not evaluated under the same conditions as the proposed ASMI-DR model. Boltz-2 and the experimental screening baselines are evaluated on restricted subsets or external protocols. This inconsistency weakens the direct comparability and may create confusion about ASMI-DR’s relative performance.
2. The paper does not clearly describe the training setup for certain baselines, particularly GNN-CLS and MAARDTI.
3. Table 1 contains several missing entries (e.g., precision/recall for SVM, AveP for some models) without any accompanying explanation.
4. The authors could discuss related work, “Predicting Dose-Response Curves with Deep Neural Networks.”





Alonso-Campaña, José, et al. “Predicting Dose-Response Curves with Deep Neural Networks.” Proceedings of the 41st International Conference on Machine Learning, 2024, https://proceedings.mlr.press/v235/alonso-campana24a.html.

**Questions:**

1. Could the authors clarify whether the train/test splits were stratified to maintain consistent ratios of active and inactive protein–molecule pairs? Given the strong class imbalance in the dataset, ensuring similar label distributions across splits is important for fair and unbiased evaluation.

2. Could the authors include early enrichment metrics, such as EF1%, in their evaluation? This would better capture how effectively the model ranks true actives near the top of the prediction list, which is particularly relevant for virtual screening scenarios.

---

> ### Author Response · Authors · 2025-11-26
>
> Thank you for carefully reading our manuscript and recognizing the value of our framework. Supported by ASMI-DR's SOTA performance in classification task on the challenging M2OR dataset, and reducing the EC50 estimation error by 40\% compared to the analogous regression architecture, we believe that our work will be an appreciated contribution to the ICLR's protein-molecule interaction modelling community. Below, we address the first weaknesses (W) and questions (Q) raised by the reviewer, and we will continue to respond to all the points.
>
> ## W1
> We appreciate the reviewer's thorough cross-check of the evaluated baselines. To make the comparison as fair as possible, we evaluated Boltz-2 on all protein-molecule pairs in M2OR dataset, and we updated the values in Table 1, Table 2, and Table 6. These novel results support the original claim that our approach, ASMI-DR, outperforms Boltz-2 affinity module by a large margin (according to the new results, RMSLE of ASMI-DR is lower by $0.385$ log units compared to Boltz-2). The results for the screening evaluation in Table 1, Table 2 and Table 6 are based on the screening wet lab experiments and the dose-response measurements. We estimated the performance based on all available protein-molecule pairs for which both the screening and dose-response experiments were performed. Therefore, it is the most accurate estimate of the screening performance we can obtain with M2OR data.
>
> ## W4
> We thank the reviewer for pointing to this interesting work, and we include a short discussion about it in the related work section. We would like to point out that this work considers a data-rich setting, as it assumes access to the responses at each concentration. While we believe that such a dataset would be highly valuable, neither of the protein-molecule interaction datasets (M2OR, DAVIS, KIBA, BindingDB) provide this level of granularity. Instead, our work can be seen as a generalization of their approach, since we only assume access to the inflection point of the curve (i.e. EC50, Kd, etc.).
>
> We would also like to point out that a major limitation of working with raw responses is heterogeneity in data sources. For example, M2OR gathers data from several different types of experiments, some of which measure change in calcium concentration (Ca2+ in M2OR "Type" column),  others measure change in cAMP (luciferase assay), and several different cell lines are utilized in the experiments. In general, in some dose–response assays, the curve does not reach a final plateau, preventing the measurement of the maximal response induced by an active compound. Thus, the responses cannot be normalized across experiments, making protocol comparisons unreliable. Therefore, although some variability exists, EC50 values are generally considered to be comparable and stable throughout the experiments.

---

> ### Author Response · Authors · 2025-12-03
> **Part 2**
>
> ## W2
> We thank the reviewer for the comment. We now clarify in the text that the reported results for GNN-CLS and MAARDTI are taken from Hladiš et al. and the original MAARDTI publication, respectively. To further strengthen the empirical evidence, we additionally include three EC50 baselines (Table 2) and experiments on two extra datasets (Table 3). In both cases, the results consistently support the state-of-the-art performance of ASMI-DR.
>
> ## W3
> We thank the reviewer for the remark on the missing values in Table 1. We refined the presentation of the results in Table 1 (see also discussion with reviewer 4Ycs).  The only missing values are for BiLSTM model for which the code is not available online. Overall, the readability of Table 1 is largely enhanced now, but the message of Table 1 remained unchanged - ASMI-DR outperforms all previous approaches on OR-molecule activity prediction.
>
> ## Q1
> We agree with the reviewer that label imbalance is a prominent property of M2OR dataset and of drug-target interaction datasets in general. For consistency, we followed the data splitting procedure from the previous work by Hladiš et al. \[5\].
>
> To analyse the consistency of splits, we check active/inactive ratio for all M2OR splits:
>
> |                    | train set          | | valid set            |  | test set             |
> |--------------------|--------------------|-|----------------------|--|----------------------|
> |         i.i.d      |    0.330   (0.007) | |    0.351     (0.034) |  |   0.314     (0.026)  |
> | Molecule--Cluster  |    0.331   (0.019) | |    0.325     (0.044) |  |   0.325     (0.094)  |
> |  Sequence--Cluster |    0.353   (0.048) | |    0.371     (0.061) |  |   0.347     (0.196)  |
> |  Molecule--Single  |    0.336   (0.020) | |    0.332     (0.038) |  |   0.318     (0.094)  |
> |   Sequnece--Single |    0.315   (0.017) | |    0.321     (0.050) |  |   0.411     (0.124)  |
>
> As can be seen, the average ratio of active and inactive pairs remained stable across different split strategies. Standard deviation of sequence split test sets is higher due to the imbalance in "broadness", i.e. in the number of active molecules per protein (See Lalis et al. \[2\]).
>
> ## Q2
> Thank you for your valuable suggestion. As in wet-lab experiments, our unified framework yields an activity decision and only predicts EC50 values for the active pairs. This prevents a fair EF1\% comparison on EC50 with the regression baselines, and such a comparison would be misleading. Instead, we report Spearman’s $\rho$ on active protein–molecule pairs to evaluate the consistency of the predicted ranking. While this is a global ranking metric rather than a top-of-list measure, ASMI-DR outperforms the regression baselines by a large margin of 38\% (absolute gain of 0.249 in Spearman’s $\rho$), and we expect this substantial improvement to carry over to the top of the ranked list.

---

### Author Response · Authors · 2025-11-26
**Revision summary and additional benchmarks**

We are grateful to the reviewers, and we would like to thank them for their work in reviewing the manuscript. The valuable insights we received will enable us to improve the clarity of the manuscript, reinforce our conclusions, and ultimately bring a valuable piece of knowledge to the ICLR community. We appreciate that all the reviewers consider our proposed framework to be interesting and recognized its strong performance on M2OR dataset in both regression and classification tasks.

To further support our claims, reviewers generally asked for additional benchmarks on other protein-molecule dataset and comparison to drug-target affinity (DTA) prediction models. To this end, we are currently running experiments on two datasets, DAVIS and BindingDB, and several state-of-the-art DTA models. We will report the results here as soon as the experiments are finished.

---

> ### Author Response · Authors · 2025-11-28
> **Performance on DTA baselines and additional benchmarks - part 1**
>
> To further challenge our proposed framework we present below the comparison against three SOTA drug-target affinity models: DTIAM \[8\], DeepDTAGen \[9\], and ProSmith \[10\] on M2OR and on two standard DTA benchmarks, DAVIS and BindingDB (Kd), which provide experimentally measured dissociation constants (Kd) for 25772 and 42234 protein-molecule pairs, respectively.
>
> To have a fair and biologically relevant evaluation between our framework and the standard regression training objective, we evaluate RMSLE and Spearman's $rho$ on correctly predicted active protein-molecule pairs. This avoids evaluation on inactive pairs, for which the EC50/Kd is usually substituted by a "dummy" high-enough constant (1 log(µM) for both DAVIS and BindingDB (Kd) in PyTDC library). For M2OR we use 2 log(mM) and for DAVIS and BinindDB we use 0.9 log(µM) as decision thresholds for regression models, which does not yield activity decision out-of-the-box.

---

> > ### Author Response · Authors · 2025-12-03
> > **Performance on DTA baselines and additional benchmarks - part 2**
> >
> > ## M2OR
> > | Name       |  | MCC                 |  | Precision     |  | RMSLE             |  | Spearman's $\rho$ |
> > |------------|--|---------------------|--|---------------|--|-------------------|--|-------------------|
> > | Boltz-2    |  | 0.108 (0.033)       |  | 0.541 (0.117) |  | 1.110 (0.037)     |  | 0.148 (0.052)     |
> > | MAARDTI    |  | 0.555               |  | 0.700         |  |                   |  |                   |
> > | GNN-CLS    |  | 0.605 (0.02)        |  | 0.689 (0.02)  |  |                   |  |                   |
> > | DTIAM      |  | *0.658* (0.029)     |  | 0.646 (0.053) |  | 1.516 (0.093)     |  | 0.363 (0.022)     |
> > | DeepDTAGen |  | 0.592 (0.043)       |  | 0.631 (0.079) |  | 1.417 (0.243)     |  | 0.224 (0.065)     |
> > | ProSmith   |  | 0.654 (0.030)       |  | 0.649 (0.041) |  | 1.402 (0.111)     |  | 0.355 (0.045)     |
> > | ASMI-Reg   |  | 0.621 (0.048)       |  | 0.692 (0.047) |  | *1.213* (0.135)   |  | *0.399* (0.122)   |
> > | ASMI-DR    |  | **0.671** (0.016)   |  | 0.773 (0.028) |  | **0.725** (0.070) |  | **0.648** (0.065) |
> >
> > ## DAVIS
> > | Model                   |  | MCC                        |  | Precision      |  |   RMSLE                 |  | Spearman's $\rho$    |
> > |-------------------------|--|----------------------------|--|----------------|--|-------------------------|--|----------------------|
> > | DTIAM                   |  | 0.566 (0.014)              |  | 0.543 (0.013)  |  | 0.736 (0.010)           |  | 0.672 (0.017)        |
> > | DeepDTAGen              |  | *0.642*  (0.014)           |  | 0.691 (0.015)  |  | 0.788 (0.017)           |  | 0.617 (0.027)        |
> > | ProSmith                |  | 0.631 (0.010)              |  | 0.619 (0.013)  |  | **0.688** (0.004)       |  | **0.700** (0.012)    |
> > | ASMI-Reg                |  | 0.601 (0.048)              |  | 0.646 (0.063)  |  | 0.772 (0.057)           |  | 0.639 (0.023)        |
> > | ASMI-DR                 |  | **0.685** (0.005)          |  | 0.772 (0.016)  |  | *0.713* (0.028)         |  | *0.696* (0.014)      |
> >
> >
> > ## BindingDB (Kd)
> > | Model                   |  | MCC                        |  | Precision      |  |   RMSLE                 |  | Spearman's $\rho$    |
> > |-------------------------|--|----------------------------|--|----------------|--|-------------------------|--|----------------------|
> > | DTIAM                   |  | 0.606 (0.026)              |  | 0.678 (0.021)  |  | *0.820* (0.020)         |  | *0.779* (0.009)      |
> > | DeepDTAGen              |  | *0.722*  (0.014)           |  | 0.827 (0.018)  |  | 0.865 (0.029)           |  | 0.748 (0.009)        |
> > | ProSmith                |  | 0.607 (0.041)              |  | 0.678 (0.033)  |  | **0.808** (0.026)       |  | **0.785** (0.013)    |
> > | ASMI-Reg                |  | 0.446 (0.231)              |  | 0.620 (0.140)  |  | 0.926 (0.116)           |  | 0.699 (0.081)        |
> > | ASMI-DR                 |  | **0.745** (0.016)          |  | 0.839 (0.018)  |  | 0.834 (0.032)           |  | 0.774 (0.013)        |
> >
> >
> > As shown in the tables, ASMI-DR achieves Kd estimation root mean square log error (RMSLE) that is statistically indistinguishable from the best-performing regression baseline ProSmith on both DAVIS and BindingDB (paired t-test p-values of $0.106$ and $0.315$, respectively), while consistently outperforming all competing methods on the activity decision metrics (MCC and precision). On BindingDB in particular, ProSmith exhibits a tendency to assign overly low Kd values even to inactive pairs, which improves its RMSLE but leads to reduced MCC and precision. In contrast, ASMI-DR produces accurate activity predictions, achieving the highest MCC on both datasets with only a marginal sacrifice in RMSLE and Spearman’s $\rho$ relative to ProSmith.
> >
> > The results of the EC50 evaluation on the M2OR dataset highlights the strong perfomrance of our approach and our architecture. Firstly, ASMI architecture trained in a standard regression setting (ASMI-Reg) outperforms all DTA baselines in RMSLE. Strikingly, relative to its regression counterpart, ASMI-DR trained in our proposed dose-response framework yields sizeable gains across all metrics, highlighting the benefit of the proposed framework. Moreover, the MCC of ASMI-DR on M2OR and DAVIS also surpasses that of dedicated classification models (Table 1 and Table A2), indicating that incorporating the dose-response training objective not only preserves affinity estimation accuracy but also leads to superior binary activity prediction.
> >
> > We believe that these results satisfactorily address the most resonating weakness of our submission, which was raised by 3 reviewers (fFHL - W1, 4Ycs - W7, W8, pBeT - W1, W2). From our point of view, the above tables significantly reinforce our claim that ASMI-DR is already practically useful and empirically strong, and that the central idea of reframing EC50 estimation as a series of binary classifications is novel, well-motivated, and well-supported by empirical evidence.

---

> > > ### Author Response · Authors · 2025-12-03
> > > **Performance on DTA baselines and additional benchmarks - references**
> > >
> > > References:
> > >
> > > \[8\] Lu et al., Dtiam: a unified framework for predicting drug-target interactions, binding affinities and drug mechanisms. Nature Communications, 2025, doi: 10.1038/s41467-025-57828-0.
> > >
> > > \[9\] Shah et al., DeepDTAGen: a multitask deep learning framework for drug-target affinity prediction and target-aware drugs generation. Nature Communications, 2025, doi: 10.1038/s41467-025-59917-6.
> > >
> > > \[10\] Kroll et al., A multimodal transformer network for protein-small molecule interactions enhances predictions of kinase inhibition and enzyme-substrate relationships. PLOS Computational Biology, 2024. doi: 10.1371/journal.pcbi.1012100.

---

### Author Response · Authors · 2025-12-03
**Rebuttal summary - part 1**

Dear Area Chair and Reviewers,

Thank you very much for taking the time to read our paper and the discussion around it, especially in this year's unprecedented circumstances. Below, we provide a concise summary of the work and of what was clarified or strengthened during the rebuttal, from our perspective as authors.


## The paper's main claim and motivation
G-protein coupled receptors (GPCRs) are the targets for nearly 30\% of all drugs on the market, but ML models still struggle in predicting main parameters for hit identification:  activity and potency (i.e. half maximal effective concentration, EC50) of GPCRs. To fill this gap, our work proposes a novel framework to simultaneously predict EC50 and activity of GPCRs, and more broadly, the affinity for protein-molecule pairs.

**The key idea is to mimic wet lab dose-response experiments and unify direct regression EC50 prediction and a single binary activity classifier** by:
- Sampling binary labels at different concentrations from a surrogate dose-response curve derived solely from the training set EC50 values.
- Train a model, ASMI-DR, to predict the probability $P(active|s,m,c)$ of a molecule $m$ activating protein $s$ conditioned on concentration $c$.
- Fit a logistic dose-response curve to these predicted probabilities across concentrations, and derive activity from the curve maximum and EC50 (or Kd) from the inflection point.

This framing:
- aligns better with how experimental data are generated, and
- treats inactive and censored samples more naturally than standard regression, which must assign arbitrary numeric values to inactives or censored measurements.

Alongside the novel framework, we also propose **a new architecture, ASMI-DR**, combining GNN for molecular representation, ESM2 and self-attention for protein representation, and cross-attention to model the concentration-dependent protein-molecule interaction.

## Main empirical contributions

- Activity prediction: **ASMI-DR outperforms current SOTA by $10$%** on activity decision prediction on a challenging M2OR dataset.
- EC50 prediction: **ASMI-DR reduces root mean square log error (RMSLE) by more than $40$%** across in-distribution and challenging out-of-distribution (OOD) splits compared to recent DTA models and regression version of ASMI architecture (ASMI-Reg).
- We also demonstrate that the **ASMI-DR outperforms costly *in vitro* primary screening** even in the search for novel active scaffolds.

Overall, our main claim is that this dose–response framing is both well-grounded and empirically beneficial not only for GPCR but also for other proteins, even with a fixed model backbone.

## What changed / was clarified in the rebuttal
Thanks to the constructive discussion with the reviewers, we added a substantial amount of experiments and clarified several key points, which significantly improved the quality of the manuscript. The results strongly support our original claims and we summarize them below.

### Comparison to recent SOTA on M2OR dataset
The main concern of the reviewers (pBeT, 4Ycs, fFHL) is the comparison to recent drug-target affinity (DTA) models. To this end we compared ASMI-DR against three recent strong DTA models: DTIAM, DeepDTAGen, and ProSmith.

- Both ASMI-DR and its architectural backbone trained in regression (ASMI-Reg) showed SOTA performance, outperforming all DTA models. In particular, **ASMI-DR outperforms the best DTA baseline ProSmith by $48$% in RMSLE**, while still achieving higher MCC.
- We re-run Boltz-2 (a model for predicting structure and affinity) on all M2OR protein-molecule pairs for a fair comparison. These novel results remained consistent with the original claim that ASMI-DR outperforms affinity and activity modules of Boltz-2.

### Evaluation on DTA benchmarks
The second main concern of the reviewers (pBeT, 4Ycs, partially fFHL) asked for stronger evidence beyond M2OR dataset. Thus, we ran ASMI-DR (and the same backbone as a regressor, ASMI-Reg) on standard benchmarks DAVIS and BindingDB (Kd).

- On both DAVIS and BindingDB, **ASMI-DR achieves RMSLE that is competitive** with the best regression baseline (ProSmith).
- At the same time, **ASMI-DR consistently provides the best activity-decision metrics (MCC and precision) across methods**.
-  On BindingDB in particular, we observed that ProSmith tends to assign overly low Kd values even to inactive pairs, which slightly improves RMSLE but degrades classification-quality metrics. ASMI-DR balances both aspects: strong activity decisions without sacrificing regression accuracy.

These results support that the approach extends beyond M2OR and is competitive with recent DTA models on standard benchmarks, even surpassing them in activity prediction.

(See next comment)

---

> ### Author Response · Authors · 2025-12-03
> **Rebuttal summary - part 2**
>
> (continued...)
>
> ### Disentangling architecture from “paradigm”
> A central conceptual question that arose is: *how much of the gain comes from the new training paradigm, versus just a better architecture?* We explicitly addressed this by comparing three variants that share the same backbone:
>     - ASMI-Prob: classifier without explicit concentration modelling or dose–response curve.
>     - ASMI-Reg: the ASMI architecture trained as a standard regressor on EC50.
>     - ASMI-DR: our full dose–response framework.
>
> On M2OR we find that ASMI-Reg already improves over prior DTA baselines in RMSLE when used as a pure regressor. **On all datasets ASMI-DR then significantly improves both EC50 RMSLE and activity metrics (MCC) over ASMI-Reg and ASMI-Prob.** We believe this provides clear evidence that the dose–response formulation itself contributes meaningfully, over and above the backbone architecture.
>
> ### Data requirements and the role of screening data
> Reviewers expressed concern that our approach might require full dose–response curves, and discussed incorporating one-point screening assays.
>
> - In the rebuttal, we clarified that **our method does not require full curves**. It only needs activity labels and an estimated inflection point (EC50/Kd), which are widely available in M2OR, BindingDB, DAVIS, ChEMBL, etc.
> - During the discussion, we emphasize that our framework provides a natural way to incorporate noisy screening samples, which can not be straightforwardly used in the standard regression training. We conclude that a future work designing more advanced, noise-aware treatment of this abundant data source is a promising direction.
>
> ### Clarification on splits, baselines, and implementation details.
>
> - We clarified the limited applicability domain of the SVM baseline and ensured the tables highlight the most relevant related works and metrics for each task.
> - We clarified our approach for the choice of hyperparameters and architecture (e.g., GNN-based molecular encoders similar to GNN-CLS).
> - We clarify that we followed the splitting strategy of the previous work of Hladiš et al., and we verified that active/inactive ratios are stable across train/validation/test in all splits, addressing concerns about potential label-distribution imbalance.
> - We also filled the missing values in Table A2 in appendix. We would like to note that we observed inconsistent results for MAARDTI on KIBA and DAVIS compared to the values reported in the original paper. Thus, to avoid confusion, we removed these rows. We emphasize that we double-checked MAARDTI results on M2OR, and we obtained similar performance to the ones published.
> - We emphasize our choice for Spearman’s $\rho$ metric over active compounds and observe that ASMI-DR improves substantially over regression baselines, indicating improved ranking of potent actives.
>
> ## On the remaining concerns of the rejecting reviewer
> Reviewer 4Ycs remained at a reject recommendation, even after acknowledging that most of its technical concerns have been addressed. His main remaining concern is not about correctness or interest of the work, but about maturity and timing - he feels that the paper, at submission time, lacked some of the now-present experiments (standard benchmarks, benefit of the training framework vs the architecture, more detailed related work).
>
> We fully understand this concern and appreciate his caution. At the same time, from our side, we would like to respectfully note that all the feasible requested experiments are now completed and support the original claims; they do not overturn or contradict any earlier result.
>
> The point that remains open is that "a dataset including concentration values should make the proposed model much more powerful". While we fully agree with the reviewer, unfortunately, there is currently no such curated dataset for protein-molecule interaction available and these experiments can not be conducted without a substantial data gathering and curation which is a large endeavour, and a stand-alone study on its own.
>
> ## Addressed concerns of the reviewers
> We believe that the borderline and positive reviewers are generally satisfied with the rebuttal and view the method as promising and relevant for the community. During the rebuttal we addressed all the main concerns: DTA and Bolt-2 baselines (pBeT, 4Ycs, fFHL), standard drug-target benchmarks (pBeT, 4Ycs, fFHL), training setup and hyperparameter choice (pBeT, fFHL), clarification (mosk, 4Ycs), splitting strategy (fFHL).
>
> Therefore, based on the above arguments and experiments, we are convinced that if the discussion continued, the reviewers would generally increase both their score and the confidence about the results of our proposed framework.
>
> (See next comment)

---

> > ### Author Response · Authors · 2025-12-03
> > **Rebuttal summary - Closing remarks**
> >
> > ## Closing remarks
> >
> > In summary, the paper:
> > - Introduces a **novel dose-response training framework** for unified activity and EC50/affinity prediction that is conceptually aligned with wet-lab practice.
> > - Demonstrates **strong performance on M2OR against SOTA baselines**, including challenging generalization regimes.
> > - Is now supported by new experiments on DAVIS and BindingDB showing that **the approach is competitive with recent DTA methods** and that the paradigm itself adds value beyond architecture.
> >
> > In conclusion, the proposed unified EC50 and activity prediction framework offers robust performance across various protein-ligand datasets. Its accuracy and versatility are likely to attract widespread attention in the community, offering strong potential for applications beyond GPCRs to a wide range of protein targets.
> >
> > # Thank you
> > Finally, we are very grateful to the reviewers and to the area chair for the time invested under difficult conditions this year. We hope this summary helps to make the state of the work and the impact of the rebuttal clearer, and we thank you sincerely for your consideration.

---

### Meta-Review · Area_Chair_kn8h · 2025-12-18

**Summary:**

Reviewers consistently find the idea of framing activity prediction as a multiple classification task, rather than directly regressing EC50 values, to be interesting and innovative. However, they raise significant concerns, including a need for deeper mechanistic insight, a substantially expanded and rigorous experimental evaluation, and major improvements to the clarity and completeness of the manuscript.

1. A key conceptual weakness is the lack of deeper insight into why the proposed classification paradigm is superior. While the method of deriving classification labels from EC50 thresholds is novel, the paper does not convincingly explain why this approach outperforms direct regression, especially since it does not incorporate additional information beyond the EC50 values used in regression. A clearer theoretical or empirical justification for the performance gains attributed to the classification framework is essential.

2. The experimental evaluation is currently insufficient to support the claims. The reviewers note that the proposed model is not competitive with a simple SVM baseline, which severely undermines its purported advancement. The evaluation must be significantly strengthened by including more baseline methods, particularly state-of-the-art DTI models, and by conducting comparisons across different model architectures to disentangle the contributions of the architecture versus the classification paradigm itself. Furthermore, the reliance on a single dataset (olfactory receptor data) limits the assessment of generalizability. Testing on additional, diverse biological activity datasets is necessary.

3. The manuscript suffers from several issues regarding clarity and reproducibility. Critical details are missing, including the training setup and hyperparameter choices for the baseline models, which prevents fair comparison and replication. Additionally, Table 1 contains missing entries that need to be completed. The writing should be polished to ensure precision and completeness throughout.

4. Finally, the related work section requires expansion and sharper analysis. A more thorough and comparative literature review is needed to properly position the contribution of this paper within the field.

**Reviewer Concerns:**

The authors have satisfactorily addressed the concerns regarding clarity (point 3) and presentation (point 4). However, the core issues raised in points 1 and 2 remain inadequately resolved in my assessment.

Specifically for point 1, while the authors have offered new hypotheses—such as the classification paradigm acting as a form of regularization or aligning better with data generation processes—these explanations remain largely speculative. To substantiate the claimed superiority of the method, more rigorous theoretical justification or targeted quantitative experiments are required to isolate and verify the mechanism of benefit attributed to the proposed idea.

Regarding point 2, while additional experiments were provided during rebuttal, significant gaps persist. First, the proposed paradigm remains primarily validated within the authors' own architectural framework. To truly disentangle the contribution of the paradigm from the architecture, it is necessary to apply this classification strategy to existing, established models and evaluate whether it consistently yields improvement. Second, the number and breadth of baseline comparisons—though expanded—still fall short of providing a comprehensive and convincing evaluation of competitiveness. Finally, the reliability of the newly added experimental results, produced under the tight timeline of the rebuttal period, may warrant further verification to ensure robustness.

Based on these considerations, I think major revisions are required to improve the paper.

**Reviewer Scores:**

I don't think the reviewers will increase their scores. As mentioned above, some concerns have been addressed, but the core issues remain unsolved.

---

### Decision · Program_Chairs · 2026-01-26

Reject